# Aerosol direct radiative effect over clouds from a synergy of OMI and MODIS reflectances

Martin de Graaf[1], L. Gijsbert Tilstra[1], and Piet Stammes[1]

[1]R&D Satellite Observations Department, Royal Netherlands Meteorological Institute (KNMI), De Bilt, The Netherlands

**Correspondence:** M. de Graaf, Royal Netherlands Meteorological Institute, R&D Satellite Observations, Utrechtseweg 297, De Bilt, The Netherlands, martin.de.graaf@knmi.nl

**Abstract.** The retrieval of geophysical parameters is increasingly dependent on synergistic use of satellite instruments. More sophisticated parameters can be retrieved and the accuracy of retrievals can be increased when more information is combined. In this paper, a synergistic application of Ozone Monitoring Instrument (OMI) on Aura and Moderate Resolution Imaging Spectroradiometer (MODIS) on Aqua Level 1B reflectances is described, enabling the retrieval of the aerosol direct radiative

effect (DRE) over clouds using the differential aerosol absorption (DAA) technique. This technique was first developed for reflectances from SCanning Imaging Absorption spectroMeter for Atmospheric CHartographY (SCIAMACHY) on Envisat, which had the unique capability of measuring contiguous radiances from the ultraviolet (UV) at 240 nm to 1750 nm in the shortwave-infrared (SWIR), at a moderate spectral resolution of 0.2 to 1.5 nm. However, the spatial resolution and global coverage of SCIAMACHY was limited, and Envisat stopped delivering data in 2012. In order to continue the DRE data

retrieval, reflectances from OMI and MODIS, flying in formation, were combined from the UV to the SWIR. This resulted in reflectances at a limited but sufficient spectral resolution, available at the OMI pixel grid, which have a much higher spatial resolution and coverage than SCIAMACHY. The combined reflectance spectra allow the retrieval of cloud microphysical parameters in the SWIR, and the subsequent retrieval of aerosol DRE over cloud scenes using the DAA technique. For liquid cloud scenes in the south-east Atlantic region with CF > 0.3, the average instantaneous aerosol DRE over clouds in June

to August 2006 was 25 $\mathrm{Wm}^{-2}$ with a standard deviation of 30 $\mathrm{Wm}^{-2}$. The maximum area-averaged instantaneous DRE from OMI/MODIS in August 2006 was $75.6 \pm 13$ $\mathrm{Wm}^{-2}$. The new aerosol DRE over clouds dataset from OMI/MODIS is compared to the SCIAMACHY dataset for the period 2006 to 2009, showing a very high correlation. The OMI/MODIS DRE dataset over the Atlantic Ocean is highly correlated to above-cloud AOT measurements from OMI and MODIS. It is related to AOT measurements over Ascension Island in 2016, showing the transport of smoke all the way from its source region in Africa

over the Atlantic to Ascension and beyond.

# 1 Introduction

The radiative effect of aerosols is one of the least certain components in global climate models (Yu et al., 2006; Forster et al., 2007). This is mainly due to the aerosol influences on clouds. Aerosols can influence e.g. cloud formation, cloud albedo and cloud life time, through their role as cloud condensation nuclei, which are called the indirect effects of aerosols (e.g. Haywood and Boucher, 2000; Lohmann and Feichter, 2005). But even the aerosol direct radiative effect (DRE), the component of aerosol radiative forcing that neglects all influences on clouds, is still poorly constrained, due to the heterogeneous distribution of aerosol sources and sinks and the influence of clouds on global observations of aerosols. In particular, the characterisation of aerosol properties in cloudy scenes has proved challenging. Locally, the aerosol DRE can be very large and dominate the radiative forcing. The understanding of aerosol effects and the influence of aerosols on clouds would be greatly advanced with daily monitoring of aerosol DRE from passive instruments with global coverage.

The derivation of aerosol DRE over clouds is generally achieved by simultaneous observations of the cloud optical thickness (COT) and aerosol optical thickness (AOT) in a cloud scene, which is challenging from satellite observations. AOT is generally small compared to COT and difficult to establish in a scene with clouds and overlying aerosols. However, in recent years several methods have been developed that separate AOT and COT. For example, active lidar measurements have been used to derive above-cloud AOT and COT, and derive the DRE from the Cloud-Aerosol Lidar with Orthogonal Polarization (CALIOP) lidar onboard the CALIPSO satellite (Chand et al., 2009). Polarimeter measurements from the Polarization and Directionality of the Earth's Reflectance (POLDER) data can be used to simultaneously derive AOT and COT in a liquid cloud scene, making use of the different effects of spherical water droplets and irregularly shaped aerosol particles on the polarisation of light (Waquet et al., 2013). Furthermore, several techniques have been developed using Moderate Resolution Imaging Spectroradiometer (MODIS) measurements (e.g. Jethva et al., 2013; Meyer et al., 2015; Sayer et al., 2016) and Ozone Monitoring Instrument (OMI) measurements (e.g. Torres et al., 2011).

The aerosol DRE can be retrieved over cloud scenes without AOT knowledge, using shortwave reflectance measurements, such as measured by the spaceborne spectrometer SCanning Imaging Absorption spectroMeter for Atmospheric CHartographY (SCIAMACHY) on the Environmental Satellite (Envisat). By determining cloud optical thickness and droplet effective radius in the SWIR, as opposed to in the visible where absorption due to aerosols can bias the cloud retrievals (Haywood et al., 2004), the aerosol effect can be determined by comparing the true cloud-aerosol scene reflectance with a modeled cloud-only scene reflectance spectrum (de Graaf et al., 2012). The spectral difference between the scene with and without aerosols is quantified by the spectral difference, which is attributed to aerosol absorption, hence the name differential aerosol absorption (DAA).

While satellite instruments have become increasingly sophisticated, measuring at higher spatial and spectral resolution and retaining global coverage in one day for most polar orbiting satellites, there is a demand for synergistic use of instruments. Space agencies have facilitated the combined use of instruments, by building instruments with complementary functionality and flying them in formation. The best example is the Afternoon constellation (A-Train), currently flying six satellites within minutes of each other, allowing near-simultaneous observation of a wide variety of parameters. Measurements from instruments in the A-Train have been used to assess the radiative effects of aerosols above clouds (e.g. Peters et al., 2011; Wilcox, 2012;

Feng and Christopher, 2015; Lacagnina et al., 2017). A number of above-cloud AOT retrievals are compared in Jethva et al. (2014) using A-Train observations.

In this paper, measurements from OMI on-board the Aura satellite and from MODIS on-board the Aqua satellite, flying in the A-Train, are combined in a different way. The (L1B) reflectance measurements are combined to create a hyperspectral reflectance spectrum and derive a new aerosol DRE product in cloud scenes using the DAA method. The DRE is derived over the south-east Atlantic Ocean during the biomass burning season. Additionally, lidar measurements from CALIOP on Calipso in the A-Train are used here to illustrate the vertical distribution of aerosols and clouds over the study area. A comparison is provided with the original retrieval of aerosol DRE over clouds using SCIAMACHY data. This paper is organized as follows: Section 2 describes the retrieval of the aerosol DRE from hyperspectral reflectance measurements in the shortwave spectrum domain using DAA. Section 3 describes the synergy of OMI and MODIS reflectances, to create a hyperspectral reflectance spectrum with sufficient spectral resolution to apply DAA. Section 4 shows the aerosol DRE over clouds in the south-east Atlantic Ocean from OMI/MODIS, compared to the aerosol DRE over clouds, derived from SCIAMACHY hyperspectral measurements from 2006 to 2009. During these years both instruments produced accurate measurements and the SCIAMACHY data from these years have been analyzed extensively in previous publications. In section 5, additional aerosol DRE data for the years 2016 and 2017 are presented. During those years aerosol-cloud interactions have been studied using aircraft measurements over the Atlantic.

## 2 Theory

### 2.1 Differential Aerosol Absorption technique

The instantaneous aerosol DRE at the top of the atmosphere (TOA) is defined as the change in net (upwelling minus downwelling) irradiance, due to the introduction of aerosols in the atmosphere. Since the downwelling radiation is simply the incoming solar radiation, and restricting the discussion to smoke aerosols for which the extinction in the longwave radiation spectrum is small, the aerosol DRE for a cloud scene is

$$\mathrm{DRE_{aer}} = \mathrm{F_{cld}^{\uparrow}} - \mathrm{F_{cld+aer}^{\uparrow}}, \tag{1}$$

where $\mathrm{F_{cld}^{\uparrow}}$ is the shortwave upwelling irradiance in an aerosol-free cloud scene, and $\mathrm{F_{cld+aer}^{\uparrow}}$ is the shortwave upwelling irradiance of the same scene with both clouds and aerosols.

The aerosol DRE over clouds is determined from shortwave hyperspectral measurements of passive imagers, using measured reflectances of cloud scenes. The Earth reflectance is defined as the quotient of the upwelling radiance $I(\lambda)$ and the downwelling solar irradiance $E_0(\lambda)$:

$$R = \frac{\pi I(\lambda)}{\mu_0 E_0(\lambda)}, \tag{2}$$

where $\mu_0$ is the cosine of the solar zenith angle $\theta_0$. If absorbing aerosols are present above the clouds, the measured scene reflectance $R(\lambda)_{\mathrm{cld+aer}}$ will deviate from an aerosol-free cloud scene reflectance $R(\lambda)_{\mathrm{cld}}$. The reflectance difference is attributed

to radiation absorption by the aerosols above the clouds, and the resulting direct radiative effect of these aerosols is quantified by integrating the reflectance difference over all wavelengths in the shortwave spectrum and all angles:

$$\text{DRE}_{\text{aer}} = \int\limits_{SW} \frac{\left(R(\lambda)_{\text{cld}} - R(\lambda)_{\text{cld + aer}}\right)\mu_0 E_0(\lambda)}{B(\lambda,\mu_0)_{\text{cld}}}\,\mathrm{d}\lambda + \epsilon, \tag{3}$$

where $R(\lambda)_{\text{cld}}$ is a simulated aerosol-free cloud reflectance, representative for the measured scene with the aerosols removed.
$B(\lambda,\mu_0)$ is the anisotropy factor of a scene, which is a measure of the angular distribution of the reflected radiation for a scene and used to determine the radiance from a uni-directional reflectance measurement. This is determined from the modeled cloud scene, and assumed to be unchanged by the aerosols over the clouds. $\epsilon$ represents all the instrument and retrieval errors of a single measurement. See section 4.3 for a derivation of Eq. 3 and a comprehensive treatment of all its components.

The aerosol DRE follows from the integration of the radiance difference between the simulated aerosol-free cloud scene and measured aerosol polluted cloud scene over the solar spectrum. The integration is over the part of the shortwave spectrum where aerosols significantly absorb radiation. In case of combined OMI and MODIS reflectances, the integration limits are from the start of OMI measurements (about 270 nm) to the first of the MODIS channels that are used to invert cloud parameters (1246 nm), where the aerosol absorption is assumed to have become negligible.

The instantaneous aerosol DRE can also be derived for cloud-free scenes (substituting $F_{\text{cld}}$ with $F_{\text{clear}}$ in Eq. 1). However, since the shortwave reflectance can be very small over dark scenes, the DAA method would produce very small numbers, yielding highly uncertain DREs. Therefore, the observations presented in this paper are restricted to cloud scenes only. Aerosol DRE for clear skies should be determined from observations of AOT in clear skies. Note that the more general all-sky direct radiative effect of aerosols in both clear and cloudy scenes is often derived as $\text{DRE}_{\text{all sky}} = f_{\text{cld}} \cdot \text{DRE}_{\text{cld}} + (1 - f_{\text{cld}}) \cdot \text{DRE}_{\text{clear}}$. (e.g. Zhang et al., 2016; Kacenelenbogen et al., 2019). Here, $\text{DRE}_{\text{cld}}$ is the direct radiative effect of all aerosols in a completely overcast atmosphere, $\text{DRE}_{\text{clear}}$ the direct radiative effect of all aerosols in a cloud-free (Rayleigh) atmosphere, and $f_{\text{cld}}$ is the fraction of clouds. However, the validity of this equation, known as the independent pixel approximation (Marshak et al., 1995; Zuidema and Evans, 1998), is dependent on pixel size and cloud homogeneity. The cloud fraction $f_{\text{cld}}$ is the fraction of an area where clouds appear with similar radiative properties. This may be true for satellites with sufficiently small pixels and homogeneous cloud fields. However, in this paper the aerosol DRE is derived from OMI, which has a relatively large footprint. For OMI an *effective* cloud fraction is derived (the OMCLDO2 product) (Veefkind et al., 2016), similar to the Fast Retrieval Scheme for Clouds from the Oxygen-A band (FRESCO) algorithm (Wang et al., 2008), but using the $O_2$-$O_2$ absorption band at 477 nm, and the DRE is derived for OMI pixels with an effective CF > 0.3 to ensure sufficiently clouded scenes. The effective cloud fraction differs from the geometric cloud fraction in that it is radiatively equivalent to the brightness of the scene but assuming a thick cloud with a fixed albedo of 0.8. The reason is that for large OMI pixels, partial cloudiness and varying optical thickness cannot be discriminated. Usually, pixels with CF > 0.3 are fully covered with clouds. Therefore, COT and cloud droplet effective radius (CER) are retrieved assuming a completely clouded scene. Then, the aerosol DRE is computed using those cloud parameters again assuming complete cloud coverage. Although this is common for satellite cloud products, it should be understood that the OMI aerosol DRE dataset is not equivalent to $\text{DRE}_{\text{cld}}$ above. A large part of the scenes with either

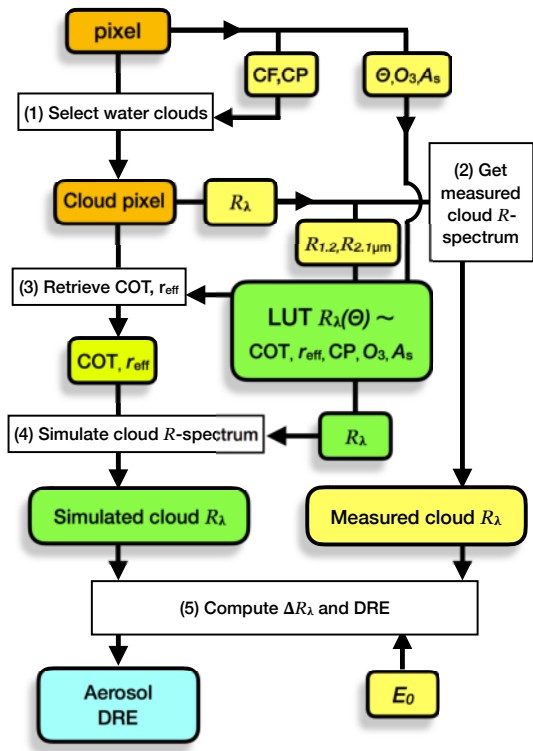

**Figure 1.** Flow diagram for the Differential Aerosol Absorption technique. Yellow boxes contain pixel products, green boxes contain simulated quantities, the yellow/green box is a retrieval for the cloud pixel and the light blue box is the end product. $\Theta$ represents the geometry of the measurements, $E_0$ is the irradiance spectrum, $R_\lambda$ is the reflectance (spectrum), CF is cloud fraction, CP is cloud pressure, COT is cloud optical thickness, $r_{\text{eff}}$ is cloud droplet effective radius, $O_3$ is the ozone profile, and $A_s$ is the surface albedo. See text for details.

small (geometrical) cloud fraction or small cloud optical thickness are not considered by selecting only scenes with effective CF > 0.3. These scenes will have a small positive or negative aerosol DRE, as aerosol scattering dominates over dark surfaces. Therefore the average OMI aerosol DRE in this paper is higher than the average true cloud or all-sky aerosol DRE. However, the dataset can be used to validate simulations of the aerosol DRE or other observational datasets where also scenes with CF

5  > 0.3 are selected. For example, the SCIAMACHY aerosol DRE over clouds was compared to HadGEM2 simulations, which showed a clear underestimation of the aerosol DRE simulated by the model (de Graaf et al., 2014). A recent comparison with POLDER aerosol DRE for pixels with a cloud fraction larger than 0.3 shows that the aerosol effect could be even higher for thick plumes (de Graaf et al., 2019). The POLDER DRE correlated very well with SCIAMACHY and OMI/MODIS DRE, but was even higher for very large values.

## 2.2 Retrieval

An illustration of the DAA technique is given in Fig. 1. The first step is the selection of suitable scenes, i.e. the selection of scenes with clouds, see above. To ensure the selection of (low level) water clouds, only pixels with a cloud pressure larger than a threshold (e.g. 800 hPa) are selected. Step two is the determination of a measured scene reflectance spectrum. For SCIAMACHY this was trivial, the combination of OMI and MODIS reflectances is treated in section 3.5. Step three is the retrieval of the cloud optical thickness and cloud droplet effective radius, using the SWIR part of the reflectance determined in step two, (e.g. $R_{1.2\mu m}$ and $R_{2.1\mu m}$). The SWIR part of the LUT of reflectances is inverted to retrieve COT and $r_{eff}$. The fourth step is the simulation of the cloud scene reflectances in the UV, visible and SWIR part of the spectrum. This forward step is simplified using the same LUT as before, which contains reflectances at 18 wavelengths from 295 nm to 2130 nm, see table 1. Once the simulated and measured cloud scene reflectances are available, the DRE is computed in step five, using Eq. 3 and a measured or reference solar irradiance spectrum $E_0(\lambda)$.

A number of alternatives steps can be identified in this scheme. Firstly, the accuracy of simulating a cloud scene reflectance spectrum can be determined by adding an extra selection criterion in step one. The Aerosol UV-Absorbing Index (AI) has been identified as a very good proxy for the presence of UV-absorbing aerosols in a (cloud) scene (e.g. Wilcox, 2012; Yu and Zhang, 2013; Alfaro-Contreras et al., 2014). By filtering for any cloud scene with a large AI, scenes with UV-absorbing aerosols above clouds are effectively filtered. If this criterion is added to step one, the remaining cloud scenes should yield a zero aerosol DRE. The (average) deviation from zero is a good estimate of the uncertainty in simulating the cloud scene reflectance. This is treated in section 4.3 for OMI/MODIS pixels. Note, however, that the exact AI threshold value is dependent on the definition of the AI, which is different for different instruments and AI products, and highly dependent on the calibration of the instrument. In the analysis in section 4.3 the OMAERO 354/388 nm AI product v1.2.3.1 was used, and it was found that a threshold of -1 was a better threshold for the removal of scenes with absorbing aerosols.

**Table 1.** Spectral cloud reflectance LookUp Table nodes

| Parameter | Nodes | | | | | | | | |
|---|---|---|---|---|---|---|---|---|---|
| wavelength $\lambda$ [nm] | 295 | 310 | 320 | 330 | 340 | 380 | 430 | 469 | 555 |
| | 610 | 645 | 858 | 867 | 1051 | 1240 | 1246 | 1640 | 2130 |
| cloud optical thickness $\tau_{cld}$ | 2 | 4 | 8 | 12 | 16 | 20 | 24 | 32 | 48 |
| droplet size $r_{eff}$ [$\mu$m] | 3 | 4 | 6 | 8 | 12 | 16 | 20 | 24 | |
| cloud base height $z_{cld}$ [km] | 0 | 1 | 4 | 8 | 12 | | | | |
| total $O_3$ column $\Omega$ [DU] | 267 | 334 | 401 | | | | | | |
| surface albedo $A_s$ | 0 | 0.5 | 1 | | | | | | |
| droplet size eff. variance $\nu_{eff}$ | 0.15 | | | | | | | | |
| number of $\theta_0, \theta, \phi - \phi_0$ | 14 | 14 | 19 | | | | | | |

Secondly, the determination of COT in step three may be replaced by more accurate retrievals. In the current set-up, COT and $r_{\text{eff}}$ are retrieved from the measured reflectance spectrum in step three. The SWIR measurements $R_{1.2\mu\text{m}}$ and $R_{2.1\mu\text{m}}$ are used to avoid biases due to absorption by aerosols, assuming that small particles do not effectively interact with radiation at those wavelengths. This works relatively well, but is also a source of uncertainty for very thick plumes and larger particles.

If unbiased cloud parameters can be obtained from other sources, e.g. from collocated dedicated cloud instruments, the DAA method may be improved, especially for thick aerosol plumes. It may even be extended to cases with desert dust above clouds, which are currently unsuitable, because large mineral particles interact with radiation at SWIR wavelengths.

## 3   Measured cloud scene reflectance spectra

### 3.1   SCIAMACHY

Originally, the DAA technique was applied to reflectance spectra from SCIAMACHY with a FRESCO effective cloud fraction larger than $0.3$. SCIAMACHY was part of the payload of the Environmental Satellite (Envisat), launched in 2002 into a polar orbit with an equator crossing time of 10:00 LT for the descending node. SCIAMACHY was designed to measure radiation in eight channels from 240 to 2380 nm at a spectral resolution of 0.2 to 1.5 nm (Bovensmann et al., 1999). The radiance was observed in two alternating modes, nadir and limb, yielding data blocks called states, approximately $960\times480$ km$^2$ in size. A

state was divided into 13 swaths. In nadir mode, SCIAMACHY produced unique contiguous reflectance spectra from 240 to 1750 nm with an optical integration time of 1 s, by co-adding. By interpolating the spectra of pixels with an integration time of 0.25 s, a swath was divided into sixteen pixels of approximately $60\times30$ km$^2$. SCIAMACHY stopped delivering data in 2012.

### 3.2   Instrument synergy using A-Train instruments

In order to continue the DRE measurements, a combination of instruments can be used to determine a contiguous reflectance

spectrum from the UV to the SWIR. A logical choice were instruments in the Afternoon constellation (A-train), which consists of several satellite platforms flying in constellation in a polar-orbiting, sun-synchronous orbit, crossing the equator in the ascending node during the local afternoon (around 13:30 LT). The purpose is to allow the instruments on-board the platforms observe the same part of the Earth within minutes of each other. The time difference between the instruments within the A-train is controlled by keeping the various satellites within control-boxes, defined as the maximum distances to which the satellites

are allowed to drift before correcting maneuvers are executed.

The main focus here is synergistic use of measurements from instruments on-board the Aqua and Aura platforms. Aqua was launched in 2002 and Aura in 2004, following Aqua by about 15 minutes. A major orbital maneuver in 2008 of Aqua decreased the distance between the Aura and Aqua control boxes to about 8 minutes. The scene that is observed by both instruments is variable to a few minutes due to the time difference between Aura and Aqua. Additionally, measurements from the CALIPSO

platform in the A-Train were used to illustrate the vertical profile of cloud scenes, but they are not necessary in the derivation of the DRE.

In addition to the combined measurements from Aura and Aqua, a lidar onboard the Cloud-Aerosol Lidar and Infrared Pathfinder Satellite Observation (CALIPSO) was used to illustrate the vertical distribution of the atmosphere. CALIPSO was launched in April 2006 and placed between Aqua and Aura. Therefore, it provides excellent collocation in time with the OMI and MODIS observations. The main payload of CALIPSO is the Cloud-Aerosol Lidar with Orthogonal Polarization (CALIOP) lidar. It provides vertically resolved backscatter profiles of the atmosphere. Here, the Level 1B attenuated backscatter at 532 nm was used, to visualize the vertical distribution of clouds and aerosols of the atmosphere sampled by OMI and MODIS. Since the CALIOP across-track swath is very small, the measurements from CALIOP are representative for the center of the OMI and MODIS swaths only. Note that CALIOP measurements are not needed for the DAA technique.

### 3.3 OMI

OMI (Levelt et al., 2006), on-board the Aura satellite, was designed to monitor trace gases in the Earth atmosphere, especially ozone. It was built as the successor to the ESA instruments GOME (Burrows et al., 1999) and SCIAMACHY, and NASA's TOMS instruments (e.g. Fleig et al., 1986; Bhartia et al., 2013). GOME and SCIAMACHY were the first space-borne hyperspectral instruments, measuring the shortwave spectrum from the ultraviolet (UV) to shortwave-infrared (SWIR) wavelength range (up till 800 nm for GOME), from which multiple trace gases, clouds and aerosol parameters can be retrieved simultaneously. OMI was designed to measure the complete spectrum from the UV to the visible wavelength range (up to 500 nm) with a high spatial resolution and daily global coverage. The optical design of OMI is different from its predecessors, which used scanning mirrors. In OMI, the incoming radiation is projected onto a two dimensional charge-coupled device (CCD). The radiation is split and mapped spectrally in one dimension of the CCD. In the other dimension, the across-track measurements are mapped. The across-track swath width is about 2600 km, resulting in a complete global coverage in one day. The spatial resolution of OMI is typically about $15 \times 23.5 \, \text{km}^2$ at nadir to about $42 \times 126 \, \text{km}^2$ for far off-nadir (56 degrees) pixels. However, the exact footprint size is complicated, which will be treated explicitly in section 3.5. Since 2008, OMI suffers from progressive degradation, especially in far off-nadir pixels, called the row anomaly.

### 3.4 MODIS

MODIS is an imaging spectroradiometer and a key instrument on-board the Terra (EOS AM) and Aqua (EOS PM) satellites (Salomonson et al., 1989). MODIS acquires data in 36 spectral bands spanning the visible and infrared. Typical application of MODIS reflectances are measurements of the surface albedo, ocean color and phytoplankton content, trace gases, clouds and aerosols, at a high spatial resolution. In this paper, only the shortwave spectral bands are used, which typically have a spatial resolution of 250 to 500 m, and a band width of about 20 to 50 nm. The spatial and spectral specifications of the MODIS bands that are used in this paper are given in Table 2.

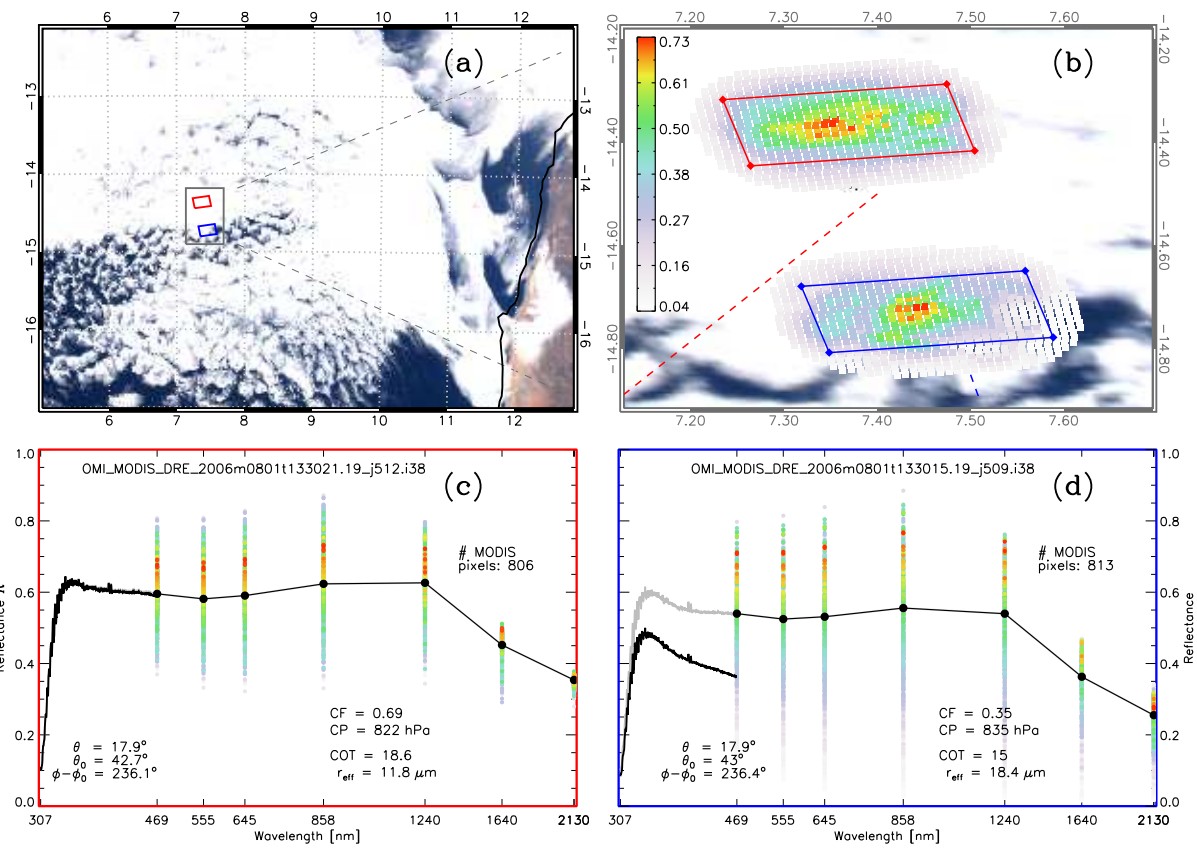

**Figure 2.** Illustration of the computation of the Aerosol DRE from a combination of one OMI pixel and collocated MODIS pixels. (a) Overview of a stratocumulus cloud deck over the south-east Atlantic Ocean using MODIS RGB, and two selected OMI pixels, in red and blue; on 1 August 2006. (b) Close-up of the two selected OMI pixels, with collocated high resolution MODIS pixels, coloured by their intensity, which is determined by the MODIS reflectance, convolved with the OMI pixel point spread function that is used to weigh the contribution of the individual MODIS pixels; (c) Shortwave spectrum from the red OMI pixel, acquired at 13:30:21 UTC, combined with the average MODIS reflectance (both in black), acquired around 13:14:15 UTC. The coloured dots indicate the weight of the individual MODIS pixels; (d) Shortwave spectrum of the blue OMI pixel, acquired at 13:30:15 UTC (black), and the average of the MODIS pixels, acquired around 13:14:09 (black). The grey curve indicates the OMI spectrum after scaling with the average MODIS spectrum. See text for details.

**Table 2.** MODIS spectral and spatial specifications of bands 1 to 7, used in this paper.

| Band | Central Wavelength [nm] | Bandwidth [nm] | Spatial resolution [m] |
|---|---|---|---|
| 3 | 469 | 459 – 479 | 500 |
| 4 | 555 | 545 – 565 | 500 |
| 1 | 645 | 620 – 670 | 250 |
| 2 | 858.5 | 841 – 876 | 250 |
| 5 | 1240 | 1230 – 1250 | 500 |
| 6 | 1640 | 1628 – 1652 | 500 |
| 7 | 2130 | 2105 – 2155 | 500 |

## 3.5 Combining OMI-MODIS reflectances

After selection of suitable cloud pixels (step one), a hyperspectral reflectance spectrum was constructed using collocated OMI and MODIS/Aqua pixels. Spectrally, OMI overlaps with MODIS at 459–479 nm (central wavelength 469 nm), which can be used to match the OMI reflectances in the visible channel and the MODIS reflectance in band 3. Spatially, the overlap is more complicated, since the OMI footprint is not uniquely defined due to the use of a polarisation scrambler. The polarisation scrambler projects four depolarized beams onto the detector CCD, which are slightly shifted with respect to each other, and therefore only the central point of the OMI footprint is uniquely defined. Furthermore, since the optics of OMI contain no moving mirror, but projects the incoming radiation onto the CCD detector array directly during a 2 s interval, the spatial response function of the OMI footprints is not box-shaped, but rather Gaussian-shaped in two dimensions. 74 % of the radiance received at a detector pixel is from within the corner coordinates, the rest of the signal is from outside the pixel corner coordinates. The OMI field of view was analyzed in detail in de Graaf et al. (2016) and Sihler et al. (2017). A 2D-Gaussian shape is used here to average MODIS reflectances across the OMI pixel, favoring pixels near the OMI center and allowing for overlapping ground pixels.

The projections of radiation are slightly different in the two OMI UV channels and the OMI visible channel, resulting in slightly different ground pixels and wavelength grids, but these have not been accounted for. All computations were performed and reported relative to the wavelength grid and ground pixels of the OMI visible channel.

Two examples of OMI pixels tiled with MODIS pixels are shown in Fig. 2. Figure 2a shows an overview of the situation: a broken cloud field over the south-east Atlantic Ocean, west of Africa, with two OMI pixels, one in the stratocumulus cloud deck (red), and one at the cloud edge (blue). Figure 2b shows the MODIS pixels that are collocated with the OMI pixels, colored by their weight in the averaging of the reflectance, which is the reflectivity convolved with the Gaussian function. Clearly, points close to the OMI pixel center are favored, but also pixels beyond the corner coordinates contribute to the radiation in the pixel. The cloud structure clearly has a large influence on the contributing pixels.

Figure 2c shows the combined OMI-MODIS reflectance of the fully cloudy scene (red), while Fig. 2d shows the combined OMI-MODIS reflectance of the broken cloud scene (blue). Clearly, there is a mismatch between OMI and MODIS for the broken cloud scene, which is caused by changes in the reflectance due to changes in the cloud fraction in the OMI footprint. The average reflectance of the scene has changed during the 15 minutes between overpasses of Aura and Aqua. The OMI/FRESCO effective CF was 0.69 in the red pixel, and 0.35 in the blue pixel. Fifteen minutes earlier, during the MODIS overpass, the geometric MODIS CF was around 0.99 and 0.98, respectively. Note that effective cloud fraction is generally lower than geometric cloud fractions. In order to get a contiguous reflectance spectrum, the average reflectance during the MODIS overpass is taken and OMI was scaled to match the MODIS average reflectance at 469 nm. Scaling MODIS to OMI seemed obvious at first, to have all parameters at the OMI grid and time. However, this resulted in very noisy data, because scaled MODIS reflectances resulted in flawed cloud parameter retrievals at longer wavelengths and the accuracy of the DRE over clouds depends strongly on the accuracy of the cloud parameters. The derivation of cloud parameters is treated below.

### 3.5.1 Cloud retrieval

In the current implementation, the MODIS reflectances at 1.2 $\mu$m and 2.1 $\mu$m are used to derive cloud droplet effective radius and cloud optical thickness, following Nakajima and King (1990) (step three). Using wavelengths in the SWIR, instead of the visible, avoids biases of cloud parameters due to absorption by overlying aerosols (Haywood et al., 2004). The cloud parameters retrieved in this way have a larger uncertainty, but can be used for scenes with overlying aerosols (de Graaf et al., 2012). Note that the MODIS reflectance at 1.6 $\mu$m is not used for the cloud retrieval, because of the large number of bad and dead pixels in the MODIS/Aqua detector (Meyer et al., 2015). The cloud droplet effective radius and cloud optical thickness are used to construct an aerosol-free cloud scene reflectance spectrum using RTM simulations ($R(\lambda)_{cld}$ in Eq. 3.) Since the retrieval of the DRE is depending so much on the correct cloud parameters and subsequent scene reflectance, the average MODIS reflectances have to be taken as a basis, and OMI reflectances have to be scaled to MODIS. The cloud optical thickness and cloud effective radii are shown in Fig. 2, representing the clouds in the two OMI pixels during MODIS overpass.

The combined, corrected reflectance spectra, as shown for the OMI pixels in Fig. 2c and 2d, are the basis for the retrieval of the aerosol DRE over clouds using Eq. 3.

## 4    Results

### 4.1    Aerosol DRE from combined OMI and MODIS reflectances

The aerosol DRE retrieval over clouds is illustrated using a case of smoke over the south-east Atlantic Ocean in August 2006. Retrieval results from both SCIAMACHY and combined OMI/MODIS measurements on 10 August 2006 are shown in Fig. 3. August is the peak of the biomass burning season in southern Africa, and an extended smoke plume, originating from the African continent, is drifting over the ocean in an elevated layer above a stratocumulus deck in the boundary layer.

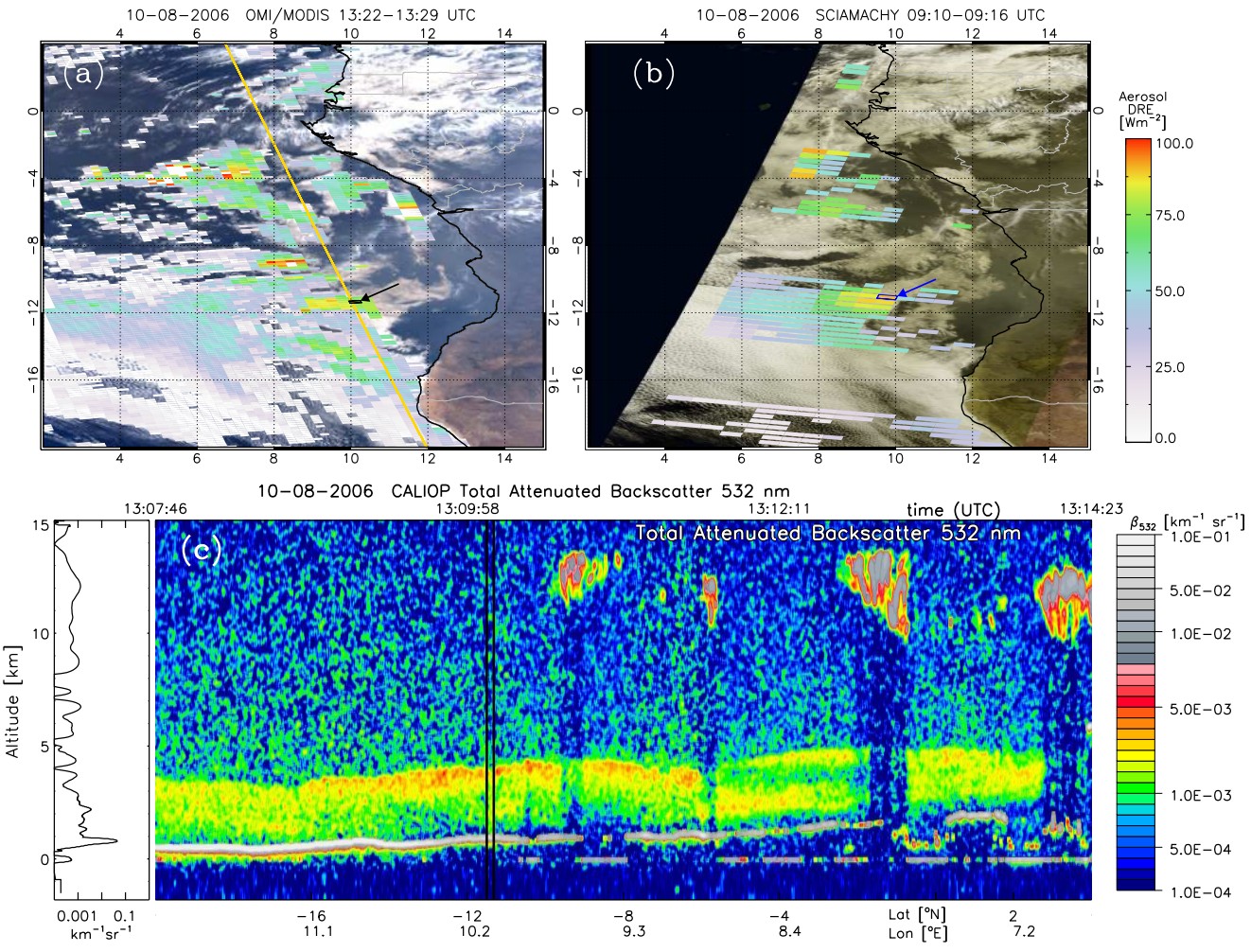

**Figure 3.** (a) Instantaneous Aerosol Direct Radiative Effect (DRE) over clouds on 10 August 2006 from a combination of OMI and MODIS reflectances, overlaid over a MODIS RGB image. The yellow line indicates the track of the backscatter profile by CALIOP that is shown in (c). The reflectance spectrum of the pixel indicated by the black arrow is given in Fig. 4; (b) Aerosol DRE over clouds from SCIAMACHY, overlaid over a MERIS RGB image. The reflectance spectrum of the pixel indicated by the blue arrow is given in Fig. 4; (c) CALIOP total attenuated backscatter at 532 nm in $km^{-1}sr^{-1}$ on 10 August 2006, for the yellow track indicated in (a). The location of the OMI pixel indicated in (a) by the arrow is indicated by the black vertical lines. The average CALIOP backscatter profile between the black lines is plotted on the left as a function of altitude.

The presence of the smoke can be observed in the RGB images of Fig. 3a and 3b as a gray haze over the continent, a darkening of the clouds and high DRE values due to absorption of radiation by smoke above the stratocumulus cloud deck.

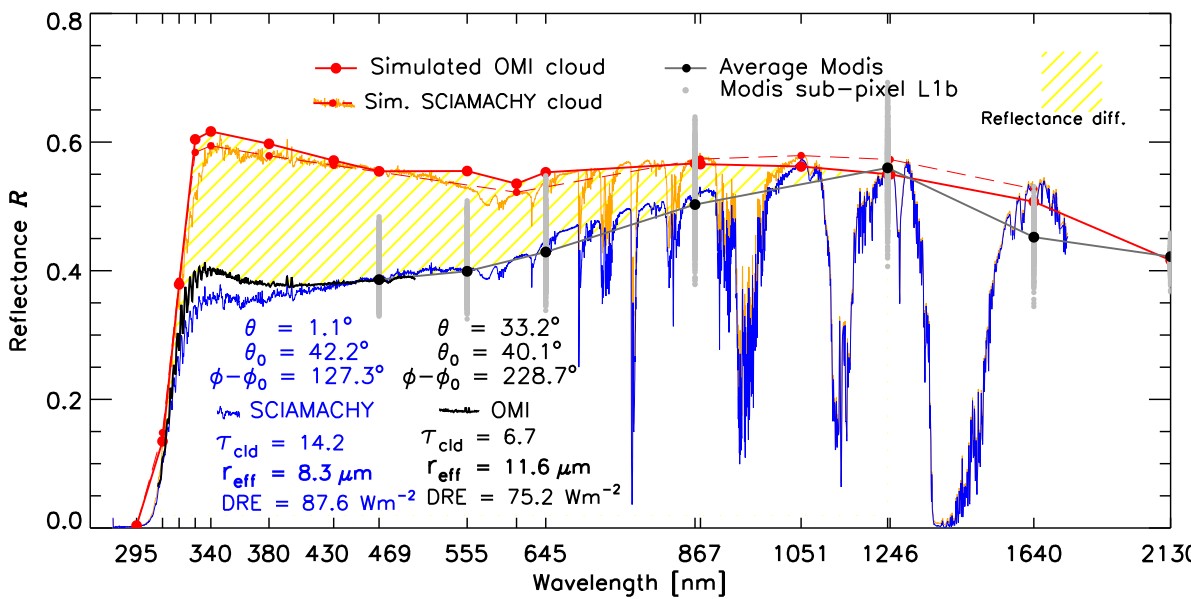

**Figure 4.** The differential Aerosol Absorption technique illustrated with OMI/MODIS and SCIAMACHY spectra. In black the spectrum measured by OMI and MODIS is given for the pixel indicated by the black arrow in Fig. 3a. In blue the SCIAMACHY measured spectrum is shown for the blue pixel in Fig. 3b. The red solid line shows the simulated aerosol-free cloud spectrum computed with an RTM for the OMI pixel. The dashed red line shows the aerosol-free cloud spectrum simulated with an RTM for the SCIAMACHY pixel.

This cloud deck is typical for this part of the ocean due to upwelling at the east part of the basin, cooling the sea surface. The stratocumulus cloud deck is persistent in the south and breaks up towards the equator.

The vertical distribution of the aerosols and clouds is illustrated in Fig. 3c, using CALIOP attenuated backscatter at 532 nm along a track shown in Fig. 3a. It clearly shows the boundary layer stratocumulus clouds between 0 and 1 km altitude, rising
towards the equator, and a thick smoke plume between 1 to 4 km altitude. The strong returns are the surface at 0 km and cirrus clouds around 12 to 14 km.

The smoke consists of small particles, which scatter and absorb the incoming sunlight. Scattering dominates and over a dark background like the ocean the planetary albedo is increased due to the smoke. This will result in a negative direct radiative effect. However, over clouds the aerosol direct radiative effect becomes positive, because the cloud optical thickness is large
and the aerosols do not contribute much to the scattering of the sunlight. They do however, absorb radiation in the visible and UV part of the shortwave spectrum, reducing the planetary albedo, resulting in a positive aerosol radiative effect over clouds. This is quantified by the OMI/MODIS aerosol DRE over clouds (Fig. 3a).

The OMI/MODIS DRE reaches values up to $100 \, \mathrm{Wm}^{-2}$ in parts where smoke from the African continent is abundant. The values drop off to zero over clouds where the smoke plume is thinning and towards the cloud edges. The high and low values
coincide well with concurrent measurements of SCIAMACHY DRE, shown in Fig. 3b. This figure shows the SCIAMACHY

DRE overlaid over a MERIS RGB image, both on Envisat. Obviously, the spatial coverage of SCIAMACHY is much lower than OMI and MODIS, measuring in nadir mode only half of the time, and having larger pixels. Consequently, the OMI/MODIS DRE is smoother with better coverage.

The location of the black OMI pixel (pointed at by the black arrow in Fig. 3a,) coincides with the blue SCIAMACHY pixel in Fig. 3b (indicated by the blue arrow) and the black lines in Fig. 3c. The computation of the DRE using the DAA technique for these pixels is illustrated in Fig. 4. The OMI reflectance spectrum up to 500 nm of the black pixel is plotted in black, complemented with the average reflectance from collocated MODIS pixels (black dots). The variations in the reflectances of the individual MODIS pixels are shown by grey dots.

The retrieved cloud droplet effective radius for this OMI scene was 11.6 $\mu$m, and the cloud optical thickness was 6.7. The aerosol-free cloud reflectance spectrum for this scene, computed with these cloud parameters (step four), is shown by the red solid line in Fig. 4. By construction, the simulated reflectances match the MODIS measured reflectances at 1.2 $\mu$m and 2.1 $\mu$m. Note that the average MODIS reflectance at 1.6 $\mu$m does not match the simulated reflectance, due to dead and bad pixels in this band.

Comparing the black and red lines in Fig. 4, differences can be observed between the simulated and measured reflectances by OMI and MODIS in the visible and UV. This is indicated by the yellow shaded area. The difference between the measured reflectance and the simulated scene reflectance is attributed to aerosol absorption by aerosols above the cloud layer in the real scene, which is not present in the simulated cloud-only scene, and used to compute the DRE following Eq. 3 (step five). The DRE derived for this OMI scene was 75.2 Wm$^{-2}$.

## 4.2 Comparison with SCIAMACHY

In the same Fig. 4, the reflectance measured by SCIAMACHY is shown in blue, for the pixel indicated by the blue arrow in Fig. 3b. This is a scene which is at the same location as the OMI pixel in Fig. 3b, but measured three hours earlier. As can be seen in the RGB images, the cloud structures have changed rather considerably during this time, but the reflectance spectra from SCIAMACHY and OMI/MODIS are still remarkably similar. The DRE was also retrieved for this scene, using the SCIAMACHY reflectances at 1.2 $\mu$m and 1.6 $\mu$m. The cloud droplet effective radius during SCIAMACHY overpass was 8.3 $\mu$m and the cloud optical thickness was 14.2. The simulated aerosol-free cloud scene reflectance spectrum for these cloud parameters is shown in Fig, 4 as the red dashed line. The SCIAMACHY DRE using the reflectance difference between the simulated cloud scene and the measured scene is 87.6 Wm$^{-2}$, which is slightly larger than observed by OMI. This is mainly due to the higher cloud optical thickness, for which the DRE is most sensitive.

The OMI/MODIS DRE is further compared with SCIAMACHY DRE over the south-east Atlantic area. SCIAMACHY has been used before to analyse the impact of smoke during the African biomass burning season on the radiation budget (e.g. de Graaf et al., 2007, 2010). Very high area-averaged instantaneous DRE were found in August 2006 of more than 80 Wm$^{-2}$, which could not be reproduced by global climate models (de Graaf et al., 2014). These high DRE values have since been confirmed by POLDER measurements (Peers et al., 2015), which show even higher instantaneous DRE values than those with SCIAMACHY (de Graaf et al., 2019). The area-average instantaneous DRE over the south-east Atlantic was also determined

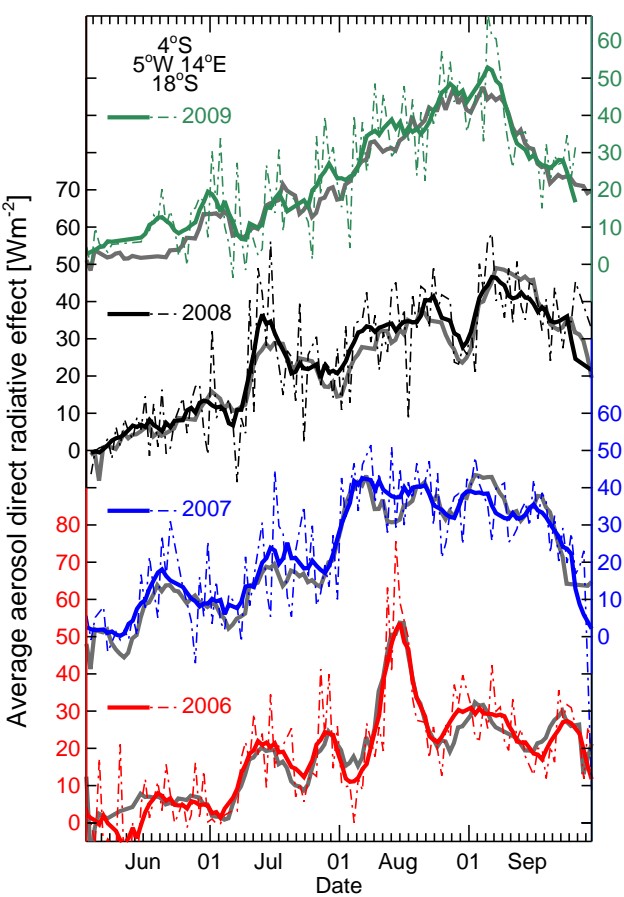

**Figure 5.** Area-averaged instantaneous aerosol DRE in Wm$^{-2}$ for the region 4 to 18° S; 5° W to 14° E (local overpass times from about 9:00–10:30 UTC) in 2006–2009 (thin lines) and its 7-day running mean (bold lines) in colored lines for all OMI/MODIS pixels with CF>0.3 and CP>800 hPa. In bold grey the SCIAMACHY area averaged aerosol DRE is plotted for CP>0.3, CP<800 hPa, which was published in de Graaf et al. (2014).

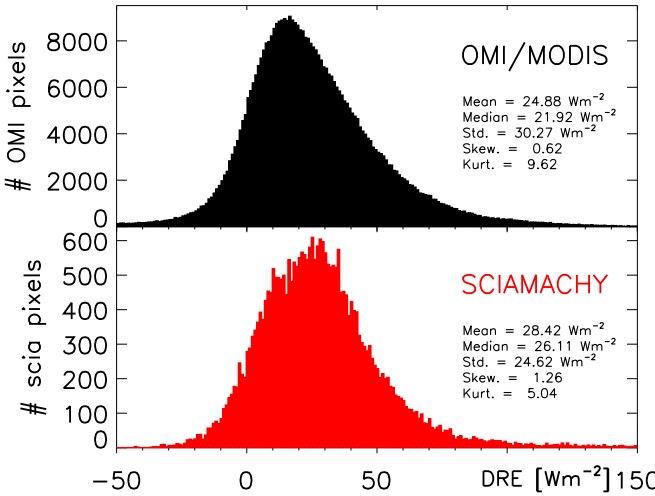

**Figure 6.** Histograms of SCIAMACHY and OMI/MODIS DRE in June to September 2006 over the south-east Atlantic Ocean (20° S to 10° N; 10° W to 20° E).

from OMI/MODIS combined reflectances, and compared to the SCIAMACHY DRE (Fig. 5). Only OMI pixels with an OMI $O_2$-$O_2$ cloud fraction (Veefkind et al., 2016) larger than $0.3$ were selected, to ensure a sufficiently clouded scene, and only OMI pixels with an $O_2$-$O_2$ cloud pressure larger than $800$ hPa, to exclude ice clouds. The maximum area-averaged instantaneous DRE from OMI/MODIS in August 2006 was $75.6 \pm 13$ Wm$^{-2}$. The SCIAMACHY data were similarly filtered, using

FRESCO cloud fraction (Wang et al., 2012) larger than $0.3$ and FRESCO cloud pressure larger than $800$ hPa. The comparison is remarkably good, considering the much better OMI spatial coverage compared to that from SCIAMACHY. Pearson's correlation coefficient for the seven-day averaged DRE values from SCIAMACHY and OMI/MODIS is 0.9667. A fit between the two datasets showed that the DRE from OMI/MODIS was about 5 % lower than that retrieved from SCIAMACHY on average with an offset of 2.4 Wm$^{-2}$.

Histograms of the DRE distribution during June to August 2006 are presented in Fig. 6. The average aerosol DRE over clouds was $25$ Wm$^{-2}$ with a standard deviation of $30$ Wm$^{-2}$ from OMI/MODIS measurements, while it was $28$ Wm$^{-2}$ with a standard deviation of $25$ Wm$^{-2}$ from SCIAMACHY measurements.

### 4.3 Accuracy assessment

In order to provide an error estimate for the OMI/MODIS DRE measurements, the uncertainty $\epsilon$ in Eq. 3 is analyzed in this
section.

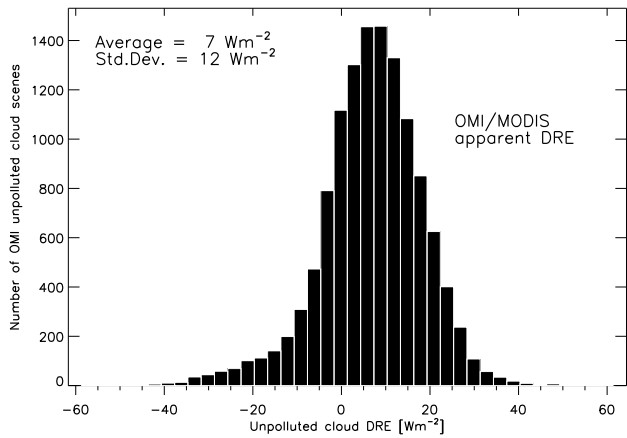

**Figure 7.** Frequency distribution of the apparent aerosol effect of all OMI aerosol-unpolluted marine water cloud scenes in June–Sept. 2006 over the south-east Atlantic Ocean (20° S to 10° N; 10° W to 20° E). The OMI/MODIS DRE for each pixel with OMI AI<0, CF>0.3 and CP>800 hPa was considered. The offset (apparent DRE) for these pixels is 7 Wm$^{-2}$, which is taken as the bias of the OMI/MODIS DRE method. The standard deviation of the DRE for these unpolluted scenes is 12 Wm$^{-2}$, which is a measure of the random error of the DRE.

### 4.3.1 Spectral cloud modeling

The most important error source is the modeling of unpolluted cloud spectra, or the ability to represent an aerosol-free cloud spectrum by a simulated spectrum. This assumption can readily be tested by comparing measured aerosol-free cloud spectra $R_{\text{cld}}^{\text{meas}}$ to simulated spectra $R_{\text{cld}}^{\text{sim}}$ for scenes that are screened for absorbing aerosols, as explained in section 2.1. The difference

$R_{\text{cld}}^{\text{meas}} - R_{\text{cld}}^{\text{sim}}$ should ideally be zero, so the resulting aerosol DRE from these scenes should be zero. Figure 7 shows the aerosol DRE for aerosol-free cloud scenes in June to August 2006. Only scene were considered with an OMI $O_2$-$O_2$ effective cloud fraction larger than 0.3 to ensure a sufficiently clouded scene, and an OMI $O_2$-$O_2$ cloud pressure higher than 800 hPa, to exclude ice clouds. To ensure the absence of absorbing aerosols an OMI with an OMAERO AI v.1.1.1 smaller than 0, following de Graaf et al. (2005). The average difference in DRE between the simulated and real scenes was about 7 Wm$^{-2}$,

was previously considered a systematic error of the differential absorption technique for aerosol-free scenes. However, the exact threshold for AI to exclude aerosols is not unambiguous, and a test with different AI thresholds showed that the average DRE for OMI/MODIS aerosol-free cloud scenes is reduced to only 1 Wm$^{-2}$ when scene with AI smaller than -1.0 are considered. Therefore, a bias due to cloud modeling may be much smaller than the 7 Wm$^{-2}$ shown in Fig. 7.

    The standard deviation for the apparent DRE between simulated and real spectra shown in Fig. 7 was 12 Wm$^{-2}$. The

standard deviation was not sensitive to a change in AI threshold, and can be considered a random error.

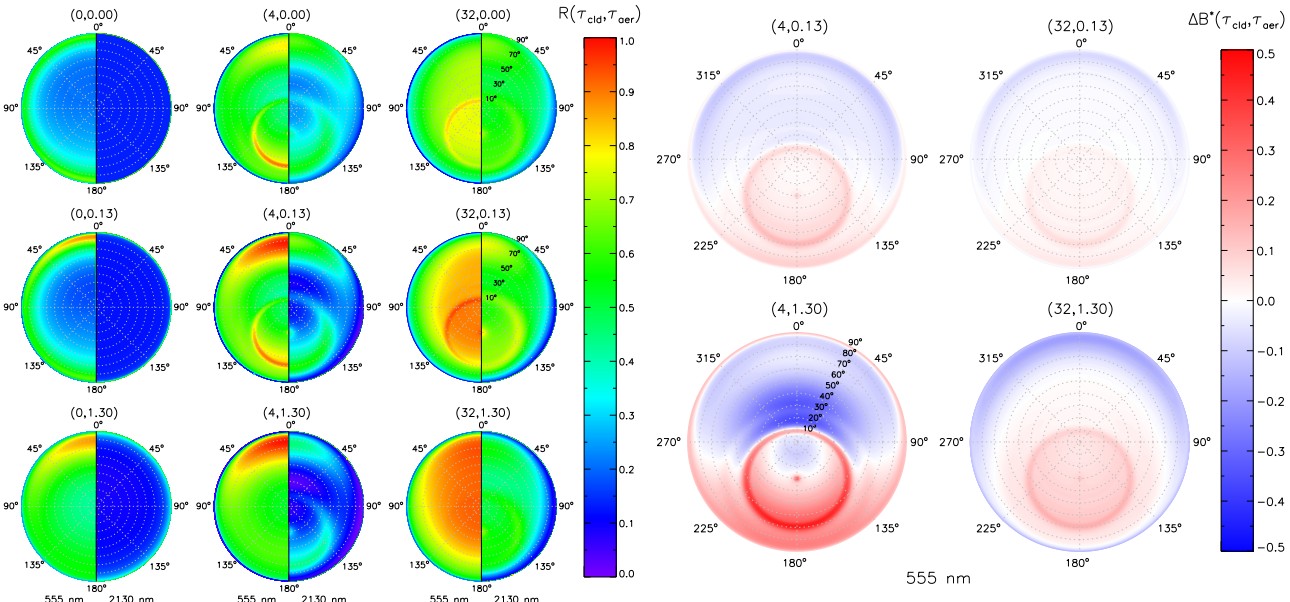

**Figure 8.** (left) Polar plot of the spectral BRDF of a scene as a function of viewing zenith angle (range of the polar plot) and relative azimuth angle ($\phi$ of the polar plot), at 555 nm (left hemisphere) and 2130 nm (right hemisphere), for different COT and AOT (given in brackets). From left to right the COT increases, while from top to bottom the AOT increases. Thus, the top-left plot represents the spectral BRDF for a Rayleigh atmosphere, while the right-bottom plot shows the spectral BRDF of an atmosphere with a cloud (COT=32) and a thick smoke layer above (AOT=1.3) at 555 and 2130 nm. (right) Spectral BRDF change $\Delta B^*$ (compared to the aerosol-free case, see Eq. 7) for the different cloud with smoke scenes, given for 555 nm. The cloud-free cases have been omitted.

### 4.3.2  Anisotropy factor

The effect of assuming an unchanged anisotropy factor between polluted and unpolluted scenes is treated in the current section, following the analysis in Prouty (2016). This thesis describes the maximum uncertainty that can be expected in aerosol direct radiative effect using Eq. 3 by simulating a cloud scene with and without (smoke) aerosols above the cloud.

5      The anisotropy factor B is defined as the Bi-directional Reflectance Distribution Function (BRDF) of a scene normalized by the spectral planetary albedo A, which is defined as

$$A(\lambda, \mu_0) = \frac{F^\uparrow(\mu_0)}{E_0(\lambda) \cdot \mu_0} = \frac{1}{\pi} \int\limits_{0}^{2\pi} \int\limits_{0}^{1} R(\lambda; \mu, \phi; \mu_0, \phi_0) \mu \mathrm{d}\mu \mathrm{d}\phi \tag{4}$$

and

$$B(\lambda, \mu_0) = \frac{R(\lambda; \mu, \phi; \mu_0, \phi_0)}{A(\lambda, \mu_0)}. \tag{5}$$

The anisotropy factor of a cloud scene is strongly dependent on scattering angle, since the BRDF of a cloud scene has some strong peaks, especially in backscatter conditions (glory) and around 140° (cloud bow). It can be shown that the uncertainty in the DRE retrieval is

$$\mathrm{DRE_{aer}} - \mathrm{DRE_{aer}^*} = \mathrm{F_{cld+aer}^{\uparrow}} \Delta B^*, \tag{6}$$

where $\mathrm{DRE_{aer}^*}$ is the DRE when the actual anisotropy factor $B_{cld+aer}$ is used instead of the aerosol-free anisotropy factor $B_{cld}$. $\Delta B^*$ is the relative difference in anisotropy factor,

$$\Delta B^*(\mu, \phi; \mu_0, \phi_0) = \frac{B_{cld}(\mu, \phi; \mu_0, \phi_0) - B_{cld+aer}(\mu, \phi; \mu_0, \phi_0)}{B_{cld}(\mu, \phi; \mu_0, \phi_0)}. \tag{7}$$

In other words, the difference between the 'true' DRE and the DRE derived assuming an unchanging anisotropy factor $B$ is proportional to the change in anisotropy factor $\Delta B^*(\lambda; \mu, \phi; \mu_0, \phi_0)$ only.

To estimate the uncertainty introduced by the assumption of an unchanging anisotropy factor, the BRDF for scenes with aerosols and clouds was simulated for different COT and AOT. For the simulations, a cloud was placed between 1 and 2 km and an aerosol layer between 2 and 5 km altitude. The clouds were simulated assuming a single-mode gamma particle size distribution with effective radius $r_{eff} = 16 \mu m$ and an effective variance $\nu_{eff} = 0.15$. For the aerosols, a bi-modal log-normal size distribution model was used, based on the 'very aged' (5 days) biomass plume found over Ascension Island during SAFARI 2000. (Haywood et al., 2003). A refractive index of $1.54 - 0.018i$ was used for all wavelengths longer than 550 nm. However, for the UV spectral region the imaginary refractive index was modified so that the absorption Ångström exponent was 2.91 in the UV, which fits satellite observations better (Jethva and Torres, 2011). The geometric radii for this haze plume used in the simulations here were $r_c = 0.255~\mu m$ and $r_f = 0.117~\mu m$ for the coarse and fine modes, with standard deviations $\sigma_c = 1.4$ and $\sigma_f = 1.25$, respectively. The fine mode number fraction was $0.9997$. These numbers are similar to the numbers used by Prouty (2016) and the same as used in de Graaf et al. (2012) to estimate the anisotropy change for SCIAMACHY DRE.

The results are summarized in Fig. 8. In the left panel the spectral BRDF is given for different scenes. The BRDF is symmetric about the 0–180° axis, but here the left side of each polar plot shows the BRDF at 555 nm, and the right side the BRDF at 2130 nm. The nine plots show the spectral BRDF for scenes with different AOT and COT, indicated by the (AOT, COT) number pairs above the figures. The COT increases from left to right from 0 to 4 and 32, while the AOT changes from top to bottom between 0, to 0.13 and 1.3. In the left-top plot the BRDF for a Rayleigh atmosphere is shown, the right-bottom plot show the BRDF for a thick cloud with a thick smoke plume.

The difference between the left side and right sides of the polar plots show that the largest geometrical dependence of the BRDF are found at smaller wavelengths. The BRDF is more pronounced for 555 nm compared to 2130 nm. Consequently,

the effect of overlying smoke aerosols on $\Delta B^*$ is small for longer wavelengths. However, at 555 nm the effect is significant. The BRDF of cloud scenes is strongly depending on the scattering angle, with a large concentration of radiation especially in the backscatter direction and at 140° degrees. When the AOT of an overlying aerosol layer increases, these strong peaks are smoothed out, and the change in $\Delta B*$ is significant. The effect is largest for a thin cloud and thick aerosol layer (COT=4, AOT=1.3).

In the right panel of Fig. 8, the change in cloud BRDF due to overlying smoke aerosols $\Delta B^*$ at 555 nm is given, for all the scenes in the left panel with aerosols and clouds (the scenes with COT=0 have been omitted). The same figures can be given at 2130 nm, but since the changes are much smaller, they are also omitted. The right panel shows again the largest change in $\Delta B^*$, and thus DRE, for a thin cloud and thick aerosol layer, for geometries in the cloud bow.

The maximum DRE change was found for this situation (COT=4, AOT=1.3, single scattering angle=140°). The DRE changed from -8.0 to 3.7 Wm$^{-2}$. This is a moderate change, smaller than the uncertainty estimated above, but due to the low COT the DRE is small, and the DRE changes sign because of the assumption of an unchanging anisotropy factor. This underlines the fact that the DAA method is valid only for sufficiently clouded scenes. Therefore, a minimum cloud fraction of 0.3 is always applied to the scenes to derive the DRE. Consequently, the derived DRE is always positive. Also note that the scattering angle of 140° is a common angle in the measurements, occurring about 40 % of the time for measurements over the south-east Atlantic during summer, so low DRE values could easily be affected by this uncertainty. Furthermore, cloud parameter retrievals can be biased in these conditions (Benas et al., 2019), but the effects are small at SWIR wavelengths (see Fig. 8) and are neglected for the cloud retrieval. $\Delta B^*$ is small for all other situations.

### 4.3.3 Accuracy

Other uncertainties are the effect of aerosol absorption on cloud fraction and cloud pressure retrievals, and the assumption of zero aerosol absorption at 1.2 microns. All these uncertainties were found to be small (de Graaf et al., 2012), in the order of about 1 Wm$^{-2}$. Here, we assume that the random errors from these error sources are similar to those for SCIAMACHY and independent, so they can be added using standard error propagation theory. This way, the uncertainty of the OMI/MODIS DRE retrievals was found to be about 13 Wm$^{-2}$, which is almost twice that for SCIAMACHY DRE. The mean reason for this decrease in accuracy is the combination of measurements from OMI and MODIS, which do not observe a scene at exactly the same time.

## 5 Application to the 2016 and 2017 biomass burning season

During the 2016 and 2017 biomass burning season, several field campaigns have been performed in the south-east Atlantic region. From May 2016 until October 2017, an ARM Mobile Facility was installed and run on Ascension Island, providing ground-based remote sensing and in situ measurements of clouds and aerosols (Zuidema et al., 2018). Also in 2016 and 2017, aircraft measurement campaigns have been carried out from Namibia, Ascension Island and São Tomé, to sample clouds and

aerosols microphysical parameters, and measure radiation (Zuidema et al., 2016). Here, the aerosol DRE over cloud from combined OMI/MODIS reflectances during these seasons are presented.

In Fig. 9a, the aerosol DRE over clouds, averaged over the south-east Atlantic Ocean, was computed using combined OMI/MODIS reflectances from 1 June to 1 October in 2016 and 2017 for pixels with a cloud fraction larger than 0.3 and
cloud pressures higher than 800 hPa. Area-averaged instantaneous DRE values are shown by the solid line, the dashed line shows a seven day running mean. It shows the evolution of smoke from vegetation fires in Africa over the ocean. In 2016, the amount of smoke is moderate in all months, except in August, when two periods of extreme pollution over the ocean can be observed. In 2017, a gradual increase of the pollution amount is observed from June onward, until it quickly diminishes halfway September. These differences can be caused by meteorological differences, controlling the transport of the smoke from
the continent to the ocean, and by differences in the amount of fires, which is in turn also determined by meteorological factors (droughts and the onset of the rain season).

Figure 9b shows the above-cloud AOT (ACA) in the same periods, derived from MODIS (Meyer et al., 2015) (solid line) and OMI (Jethva et al., 2013) (dashed line) measurements. The correlation between the above cloud AOT and aerosol DRE over clouds is very large, especially for the MODIS ACA. Although the aerosol DRE is mainly determined by the cloud reflectance
of the cloud underneath the clouds, the correlation can be explained by the persistence of the marine boundary layer clouds over the Atlantic. These clouds are very stable, and the change in cloud fraction is small when averaged over the considered area. The large peaks in August 2006 are also visible in the ACA data, and are clearly caused by the presence of smoke.

The high values of the aerosol DRE and ACA in August and September 2016 are also reflected in AOT data collected by the AERosol RObotic NETwork (AERONET) station on Ascension Island, located at 8° S, 14.4° W. The version 2 (V2) level 1.5
AOT at 500 nm over Ascension Island from 1 June to 1 October 2016 and 2017 is shown in Fig. 9b. It shows AOT higher than 0.2 in a few isolated events in August 2016, which were strongly correlated with episodes of high aerosol DRE over clouds in the south-east Atlantic, as shown in Fig. 9a. On the other hand, in 2017 the aerosol DRE values were more moderate, and do not correlate clearly with the AOT over Ascension Island. Note that version 3 (V3) data are also available (Giles et al., 2019), but the level 1.5 AOT data showed rather different behavior than the V2 data, and the V2 data were retained. Level 2.0 data
were also available for 2016, but these are almost equal to the level 1.5 data, and for 2017 the level 2.0 data were not yet available. Therefore V2 level 1.5 data were used in Fig. 9c.

The peaks in AOT over Ascension Island lag the peaks in DRE and ACA over the Atlantic by two days. This is shown for 7 August 2016 (vertical line in Fig. 9) and in Fig. 10, which presents the aerosol DRE from OMI/MODIS during three days: 5, 6, and 7 August 2016. On the first day the aerosol DRE and AOT over the Atlantic Ocean peak (Fig. 9a and b), while during
the last day the AOT over Ascension peaks (Fig. 9c).

In Fig. 10, HYSPLIT backtrajectories (Rolph et al., 2017) of air parcels ending over Ascension Island at 500, 1500, and 3000 m altitude are overlaid over each image (same trajectories in all images). They show the rapid transport of smoke over the Atlantic originating from Angola and its back-country. The colored stars indicate the time of satellite overpass in each backtrajectory, which is around 13 UTC. On August 5 this is indicated by the brown stars, on August 6 by orange stars, while
on August 7 this is at Ascension, indicated by the yellow star.

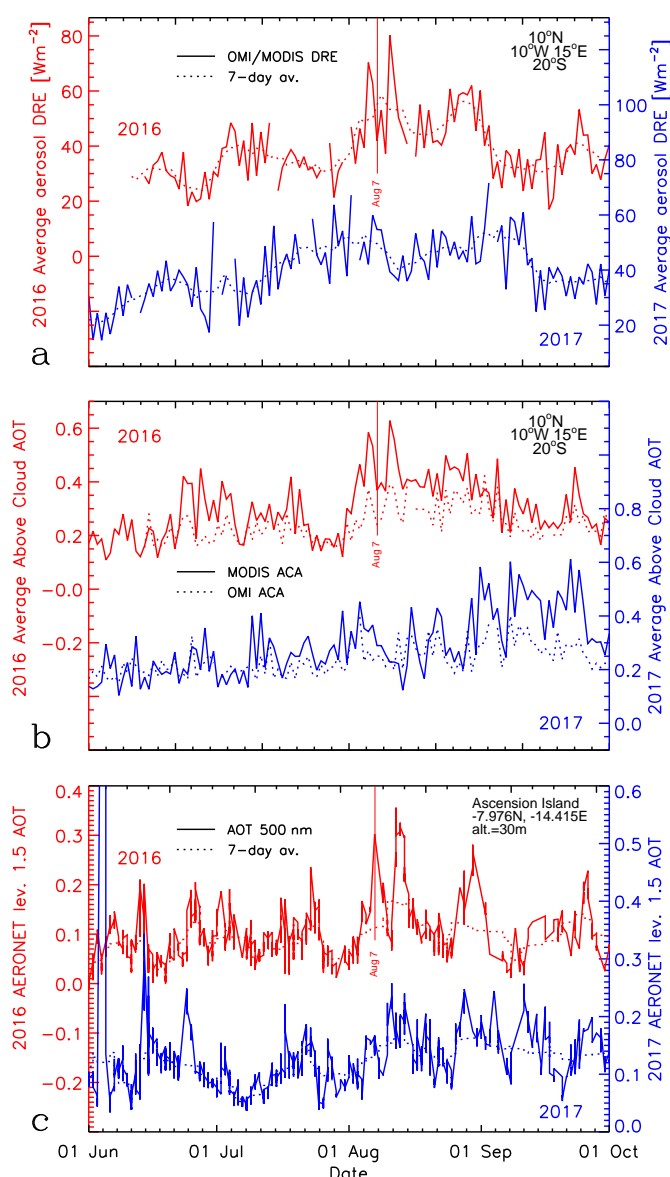

**Figure 9.** (a) OMI/MODIS aerosol DRE over clouds, averaged over the Atlantic Ocean (10° N to 20° S; 10° W to 15° E) in 2016 (red) and 2017 (blue). The solid line shows the area-average instantaneous DRE, the dashed line shows a 7-day running mean; (b) Above-cloud AOT (ACA) derived from MODIS (solid line) and OMI (dashed line) measurements during 2016 (red) and 2017 (blue), averaged over the same area as (a); (c): AERONET AOT at 500 nm from Ascension Island station at 7.98° S, 14.42° W in 2016 (red) and 2017 (blue). The solid line shows all available level 1.5 data, the dashed line shows a 100 point running mean.

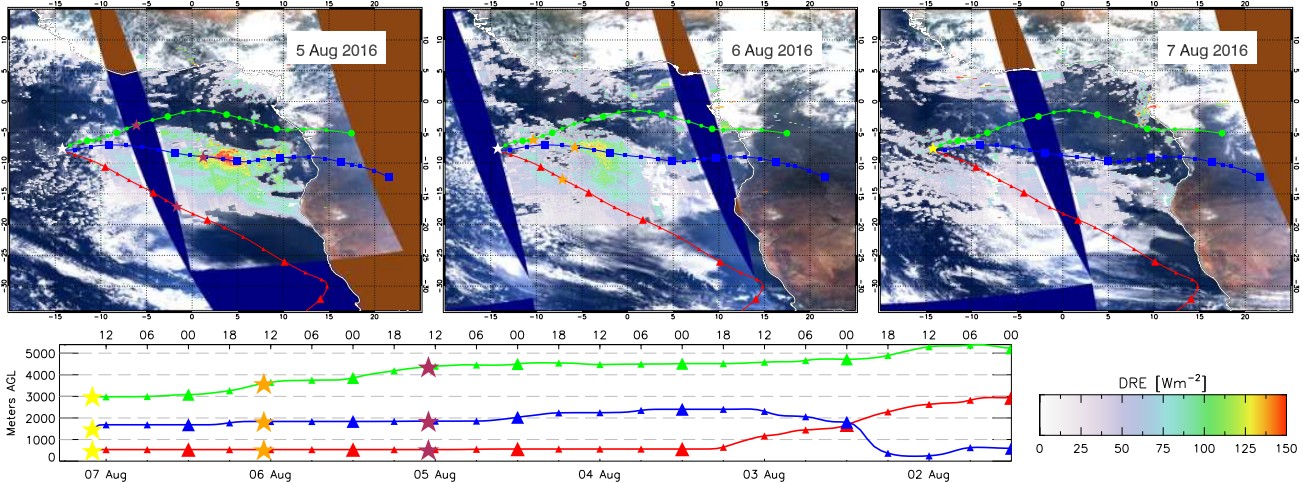

**Figure 10.** Aerosol DRE over clouds from OMI/MODIS overlaid over MODIS RGB images for three consecutive days (5,6,7 August), and backtrajectories from Ascension Island of air parcels ending at 500 m (red), 1500 m (blue) and 3000 m (green). The position of the air in the backtrajectories during the satellite overpasses is indicated by the coloured stars (yellow on 7 Aug. (at Ascension Island), orange on 6 Aug. and brown on 5 Aug.)

The wind direction in the boundary layer (500 m, red) is south-east, which is very persistent for this area. The air ending at 1500 m (blue) originates from Angola and beyond, while the air at 3000 m (green) originated somewhere around Congo. All three layers can carry aerosols and contribute to the high AOT at Ascension Island.

The boundary layer will likely contain marine aerosols, but the transport in this layer is very constant, adding to the back-
5  ground AOT over Ascension of about $0.1 - 0.2$. Only the 1500 m layer coincides exactly with the peak DRE over the ocean during 5 and 6 August, as shown by the stars in the different panels. High values of DRE travel along the blue 1500 m line, crossing the Atlantic in only a few days. Interestingly, the altitude of this layer (shown in the bottom layer of Fig. 10) is close to the ground over the continent, quickly rising to above 2000 m at some point and then gradually declining to 1500 m. This strongly suggests that the layer is smoke filled and heated over a fire area, which then travels over the ocean in a stable elevated
10  layer, as found by Swap et al. (1996). Lastly, the layer ending at 3000 m is at a high altitude at all times, and is not collocated with high DRE values, and therefore it is less likely that this layer contributes to the high AOT over Ascension.

## 6  Conclusions

In this paper, the aerosol direct radiative effect product is presented retrieved from combined level 1B reflectance measurements from OMI and MODIS. The synergistic use of multiple instruments was made possible because the instruments fly in formation

in the A-train. This presents opportunities which are not otherwise possible, or with a much lower coverage, depending on the collocation of instruments.

The aerosol DRE over clouds can be retrieved from combined OMI/MODIS reflectance spectra using the DAA technique, as it was also done using SCIAMACHY spectra. MODIS reflectance collocated with OMI pixels were used to retrieve cloud properties of a cloud scene, while the combined OMI and MODIS shortwave reflectance spectrum provides information about the absorption by aerosols in the UV and visible part of the spectrum.

This yields aerosol DRE over clouds which were compared with existing data from SCIAMACHY, using cloud scenes over the Atlantic Ocean. This area is known for its strong pollution by smoke during the south African biomass burning season, and can be used to demonstrate the strong aerosol DRE over clouds. For liquid cloud scenes with CF > 0.3, the average aerosol DRE over clouds in June to August 2006 was 25 $Wm^{-2}$ with a standard deviation of 30 $Wm^{-2}$. The maximum area-averaged instantaneous DRE from OMI/MODIS in August 2006 was $75.6 \pm 13$ $Wm^{-2}$. The OMI/MODIS DRE shows a very good correlation with SCIAMACHY DRE between 2006 and 2009, and has a much better resolution and coverage. Furthermore, SCIAMACHY stopped delivering data in 2012, while OMI and MODIS are still producing high quality data.

The successful combination of OMI and MODIS reflectances demonstrates the possibility for synergistic use of other instruments as well, other than combining L2 products. For example, the aerosol DRE over clouds may also be derived from combined Visible Infrared Radiometer Suite (VIIRS) and Ozone Mapping and Profiler Suite (OMPS) data, which could complement the current OMI/MODIS DRE dataset and that derived with SCIAMACHY, especially since OMI shows progressive instrumental degradation. These instruments both fly on the Suomi–NPP (SNPP) spacecraft since 2011, so the collocation will be much better than between OMI and MODIS. In 2017, another set of VIIRS and OMPS instruments was launched onboard the NOAA20 platform, leading SNPP by 50 minutes. More identical instruments are planned on NOAA's Joint Polar Satellite System (JPSS) program, enabling data generation for the next two decades. Furthermore, the instrument capabilities continue to grow, so the DRE may be retrieved with higher accuracy at at higher spectral and spatial resolution.

OMI/MODIS DRE data in 2016 and 2017 show the effect of smoke being transported over the Atlantic all the way to Ascension, 3000 km from its source, where it coincides with high AOT values measured by AERONET. A high correlation of the aerosol DRE over clouds was found with above-cloud AOT, even though the DRE is more strongly dependent on COT then AOT. This can be explained by the persistence of the marine boundary layer cloud deck over the south-east Atlantic. Backtrajectories show that the altitude of the smoke layer was well above the boundary layer in the free troposphere, as found by several studies before. The OMI/MODIS DRE can be used to study the aerosol direct effect, but also contribute to understanding more complex feedback mechanisms between clouds, aerosols and radiation.

*Data availability.* The OMI/MODIS DRE data are freely available on *www.temis.nl/climate/adre.html* and from the first author on request.

*Author contributions.* MdG developed the DAA technique, its application to OMI/MODIS measurements and created the SCIAMACHY and OMI/MODIS DRE datasets. LGT provided support for the satellite retrievals and developed surface reflectance datasets. PS developed the RTM.

*Competing interests.* The authors declare that they have no conflict of interest.

5   *Acknowledgements.* This work was funded by the Dutch National Programme for Space Research User of the Netherlands Space Office (NSO), project number ALW-GO/12-32. Brent Holben is thanked as PI of the AERONET station at Ascension Island and providing the AOT data. The reviewers and editor are thanked for their constructive comments and contributions to the original manuscript.

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
