# Peer review of "Aerosol direct radiative effect over clouds from a synergy of OMI and MODIS reflectances"

_Atmospheric Measurement Techniques, 2019_

## Referee Comment (RC1) · Anonymous Referee #2 · 14 Mar 2019

This paper describes a method to estimate the direct radiative effect (DRE) of aerosols above clouds using OMI and MODIS measurements. The technique (differential aerosol absorption, DAA) is somewhat different from related algorithms as is in essence a retrieval of the radiative effect itself rather than being focused on the optical/microphysical quantities of the aerosols and clouds, which gives it a different set of strengths and weaknesses from other above-cloud aerosol algorithms. The DAA retrieval is an extension of an algorithm published by the authors previously, which used SCIAMACHY instead. The SCIAMACHY record ended in 2012, while OMI/MODIS are still flying, and other sensors with similar capabilities also fly now and are planned for the future (e.g. OMPS/VIIRS, and PACE OCI). SCIAMACHY was a spectrometer with a coarse footprint, while OMI is a UV-vis spectroradiometer and MODIS is multispectral.

[Figure]

MODIS and OMI also have different footprints (both finer than SCIAMACHY) and fly on different platforms. So, the adaptation of the algorithm from SCIAMACHY to these other sensors is of scientific interest and sufficiently non-trivial and novel. This work is well in scope for AMT.

The quality of language is good. With the exception of Section 5, which was a let-down, the paper is pretty good. Overall I recommend publication after minor revisions; some points in the text need expanding and I have a few concerns with the error budget, as well as the lack of use of ORACLES data. I would be happy to review the revision.

Page 2, lines 23-27: While POLDER is probably the most informative, there are several techniques to estimate above-cloud AOT and COT from MODIS and/or OMI alone. See e.g. work by Meyer, Sayer, Jethva for various algorithms. I'm not saying that the authors have to cite each paper in this field, but a brief acknowledgment/discussion of the fact that there are several MODIS or OMI techniques which have been developed and used successfully already, and it's not only POLDER and CALIOP which have these capabilities, would be welcome.

Page 2, line 26: This mentions that a comparison with POLDER results is presented elsewhere. I went to the bibliography and this is listed as a study in preparation for submission to GRL. If this work has already been done, it would be good to briefly summarize the results. This is relevant because the POLDER technique is quite different from DAA. Otherwise, I'd just say that the comparison will be performed and remove the citation. I suppose the progress of both papers can be assessed at the time that this manuscript is revised. Given this paper is cited again on page 14, I think it's important that we get to see the results, which we can't because the paper being cited hasn't even been submitted yet. Basically, either give us the information or remove the citation.

Page 3, line 5: I'd add a brief discussion of and references in support of the assumption of negligible aerosol effects in the longwave. While agree it is probably the case for

smoke, it may not be for dust. I know there are various papers looking at shortwave vs. radiative effects of dust under various conditions (e.g. over land, ocean, daytime, nightime, cloud). I think it's important to acknowledge when/where this assumption is reasonable and the magnitude of the error from assuming it is negligible. Some readers might otherwise assume it is always negligible. This is mentioned later on page 5, but I'd state it here too.

Equation 3: this is the core of the method; the most questionable assumption here seems to me to be that the anisotropy factor B is the same for an aerosol-laden and an aerosol-free cloud. Intuitively one would expect the aerosol-laden scene to be less anisotropic. Page 4 directs the reader to de Graaf (2012), and I found that their Section 6.2 addresses this. I realise that these errors are often AOT-dependent but to give the reader a rough idea of expected performance for the SCIAMACHY case (as a reference for the present MODIS/OMI), I suggest summarizing this information here (either the total figure of 8 Wm-2 given in section 6.3 of the 2012 paper, or a brief quantification of the individual components) so the reader does not need to dig out the previous paper.

Section 4.3: If I understand this correctly, the biggest contribution to the retrieval error is estimated as the calculated forcing for pixels where the UV aerosol index (UVAI) is less than 0. This has mean and standard deviation 7 and 12 Wm-2 respectively. This is fine in theory but I have some questions in practice. UVAI is a semi-quantitative detection since it depends on not only aerosol absorption but also on factors including solar/view geometry, altitude, cloud properties, underlying surface (in cases of broken cloud) etc. The threshold value of 0 is not supported by radiative transfer arguments as far as I can tell, but rather seems a hand-waving threshold that is a nice round number. While sensible as a first approximation it is certainly possible to get negative UVAI when there is some absorption (this is even shown in the de Graaf 2005 paper the authors cite at this point), or positive when the aerosols are only scattering; while one might argue that this would contribute to the scatter in Figure 6, there is no reason to assume that it would lead to an unbiased estimate. Thus the reported systematic bias of 7 W m-2

might be true, or might be the result of choosing 0 as the UVAI threshold when another one would be more appropriate. It is not clear which UVAI the authors are using (there are several definitions and data versions). I believe the latest OMI standard product version includes a new definition and calculation which reduces the dependence on factors like geometry (see Torres et al 2018, https://doi.org/10.5194/amt-11-2701-2018 ). If this was not what was used, I recommend repeating this analysis with it. The new OMI UVAI will reduce some of these confounding effects such that it is a better proxy for aerosol absorption. It should make the authors' assessment of systematic/random errors here more realistic. So, my suggestions are: (1) Ensure that the latest OMI UVAI data set is used for this calculation, to decrease the confounding non-aerosol effects, and (2) acknowledge that UVAI=0 as a threshold is arbitrary and mention (or even better), estimate the additional uncertainty this is contributing to the error analysis in section 4.3. Perhaps a better threshold than UVAI=0 could be determined and adopted.

Section 5: Honestly this section is a bit of a let down and missed opportunity. The authors show time series of radiative effect during the CLARIFY, ORA-CLES, and LASIC campaigns, and give citations about them. However the analysis amounts to plotting back-trajectories and showing tmie series of AERONET AOT against DRE. None of the actual data from the field campaigns appears to have been actually used. The ORACLES data are already freely available from https://espoarchive.nasa.gov/archive/browse/oracles . This includes a large number of relevant observations including e.g. irradiance/flux which could be used to evaluate the algorithm's output more quantitatively, rather than just showing that AOT at Ascension Island is correlated with DRE over the southern Atlantic Ocean. I strongly urge the authors to look at these data as there are bound to be some matches close in space/time to the A-Train overpass. It would help give a sense of whether the DRE magnitudes are reasonable, as right now all we can say is that temporal variation seems reasonable. As-is, the paper's introduction and section 5 state these plots are presented "in support of" these campaigns, but there's really no linkage demonstrated in what's actually contained in the paper.

[Figure]

Conclusions: this quotes mean and standard deviations of DRE. I'd be interested to see some pdfs somewhere in the paper, to see what the distributions look like at different scales. If they are skewed then mean and standard deviation might not be the best summary metrics, perhaps median and interquartile range would be better. This could also be something to add to the SCIAMACHY comparison section, for example: show whether the pdfs of DRE are similar to within the expected level of consistency for e.g. a season's worth of data over the south Atlantic. This would complement the existing instantaneous consistency assessment with a more climatological consistency assessment, which is after all important if the end goal is to move toward a long-term post-SCIAMACHY record.

I was also surprised not to see any mention of VIIRS/OMPS in the paper. These sensors fly on SNPP (since 2011 – there's even a brief overlap with the SCIAMACHY record) and NOAA20 (since 2017), and have similar capabilities overall to MODIS/OMI. In some senses they would even be a better choice than the MODIS/OMI pair, because they fly on the same satellite, which simplifies some of the collocation/time difference issues. Again, I don't expect the authors to demonstrate the algorithm with VIIRS/OMPS, but a brief mention that this sensor combination exists and the relative merits of the sensor pair would be welcome.

---

## Referee Comment (RC2) · Zhibo Zhang (Referee) · 19 Apr 2019

Summary: This paper documents a method called differential aerosol absorption (DAA) to estimate the direct radiative effect (DRE) by the smoke aerosols above cloud (AAC) in the SE Atlantic region using the combination of OMI and MODIS. In this paper, the physical basis of this method is illustrated using selected cases, the uncertainties are analyzed. Applying this method to Aug. 2006 yields an "average aerosol DRE" of 31.5 Wm-2. The topic of this paper is a good match of AMT. The DAA method described in this paper is unique and interesting, although most aspects of this method have already described in early studies i.e., de Graaf et al. (2012, 2014). Overall, I think this paper can eventually be accepted for publication in AMT, but not without significant revision. Below is a list of my major questions and comments. They need to be addressed

carefully and thoroughly so the revised paper can meet the standards for publication in AMT.

Major concerns/questions:

1) Uncertainties associated with the anisotropy factor B: Eq. (3) is the main theoretical foundation of the DAA method. I think it needs to be explained better than what is in the current manuscript. A main uncertainty I can see is the anisotropy factor B, which is basically the angular distribution model (ADM) used by the CERES to convert the directional radiance to hemispheric flux. In this study it is assumed that the anisotropy factor for AAC is the same as that for clean clouds. But this assumption is not justified or discussed in depth in the paper. It is simply stated that the uncertainty associated with this factor was investigated in de Graaf et al. (2012). Of course, this is not satisfying and sufficient. This uncertainty needs to be carefully addressed in the present study. In particular, the following questions need to be clarified with proper figures, data and/or references

a. Anisotropy factor is a strong function of solar-satellite viewing geometry. The uncertainty can be especially large over the special scattering angles, such as rainbow directions. A figure is needed to show the difference between the anisotropy factors for AAC and for clean clouds as a function of satellite viewing direction (i.e., polar contour). This figure can be plotted using the typical solar zenith angle in July or August in the SE Atlantic region at the A-Train crossing time (i.e., 1:30 PM).

b. Moreover, the anisotropy factor for AAC is also dependent on the scattering properties of the aerosol. It has to be explained why simply assuming the same anisotropy factor B for all types of AAC is sufficient.

c. Similarly, the anisotropy factor for AAC is also dependent special wavelength. The spectral difference between the B for AAC and clean clouds also needs to be addressed and the uncertainty assessed.

2) The difference between OMI and MODIS cloud reflectance: On page 9 line 20, it is found that "Clearly there is a mismatch between OMI and MODIS for the broken cloud scene, which is caused by rapid cloud changes. The averaged reflectance of the scene has changed during the 15 minutes between overpasses Aura and Aqua." First, I found the difference between the OMI and MODIS cloud reflectance surprisingly large (i.e., ∼0.6 MODIS vs. ∼0.4 OMI). So this seems to be a Second, I found the speculation that this difference is caused by "rapid cloud changes" not convincing. Note that the underlying clouds below AAC in SE Atlantic region are mostly boundary stratocumulus clouds. These clouds are pretty stable. It is hard to imagine cloud reflectance changes 50% in only 15 minutes. Min and Zhang (2014) studied the influence of cloud diurnal cycle on the DRE estimation on the basis of MODIS and SEVIRI observations. They found about 5% cloud fraction change in the SE Atlantic region between Terra (10:30 AM) and Aqua (1:30 AM). The two satellites are separated by 3 hours and the cloud fraction change is only 5%. I am not convinced that within 15 minutes the cloud reflectance can change 50% (for convective clouds maybe, but not for stratocumulus). So this issue/question needs to be addressed and clarified with substantial evidences. I'd suggest the author to use the high-frequency SEVIRI data (∼15 minutes) to assess how much the cloud property changes within 15 minutes in the studied region.

3) Sampling rate of the DRE needs to be provided: In my opinion, the DRE values are only meaningful and useful when the corresponding sampling rate is given side by side. It seems to me that the DAA method described here is only applicable to certain portion of the total cloud fraction. But the paper provides no data or analysis of the sampling rate. As shown in Zhang et al. (2016) as well as many previous studies, the DRE is dependent on both AOT and COT (See their Figure 9a). If a method only samples, say large COT and large AOT, then the DRE from such method would yield larger DRE than another method that can sample all COT and AOT. But the results from the two methods are not directly comparable. Because of the lack of sampling rate information, it will be difficult for other researchers to compare or use the DRE results from this study. To address this issue, I believe the following sampling rates

need to be provided with proper figures or tables:

a. What is the total cloud fraction identified by the collocated OMI-MODIS observations?

b. What is the fraction of cloud with detectable AAC, e.g., UV-AI >2?

c. What is the fraction of the cloud with valid DRE estimation using the DAA method?

d. The above information should be provided for all DRE results, for example, Figure 5. I am wondering to what extent the inter-annual difference in Figure 5 is due to real cloud or aerosol and to what extent it is actually due to year-to-year sampling rate difference.

e. Another relevant question is: What are the DREs for the cloud with detectable AAC but the DAA method cannot be used for any reasons?

The sampling rates need to be provided whenever the DRE values are given, i.e., in the abstract and conclusion.

4) The DRE results need to be presented more carefully: It needs to be empathized in the abstract and conclusion that the DRE from this study is the instantaneous DRE only at the A-Train overpassing time. When talking about the "daily averaged" DRE, the following questions need to be clarified: a. Is it a diurnal average, i.e., including nighttime, or only daytime average? See Zhang et al. (2016) about diurnal average. b. Does the daily average account for the diurnal cycle cloud clouds? See Min and Zhang (2014) about the impact for cloud diurnal cycle on DRE estimation.

Min, M., and Z. Zhang (2014), On the influence of cloud fraction diurnal cycle and sub-grid cloud optical thickness variability on all-sky direct aerosol radiative forcing, Journal of Quantitative Spectroscopy and Radiative Transfer, 142 IS -, 25–36, doi:10.1016/j.jqsrt.2014.03.014.

Zhang, Z., K. Meyer, H. Yu, S. Platnick, P. Colarco, Z. Liu, and L. Oreopoulos (2016),

[Figure]

Shortwave direct radiative effects of above-cloud aerosols over global oceans derived from 8 years of CALIOP and MODIS observations, ACP, 16(5), 2877–2900, doi:10.5194/acpd-15-26357-2015.

---

## Editor Comment (EC1) · Hiren Jethva (Editor) · 6 May 2019

Dear Author,

The open review period of your submitted manuscript is now closed. Your manuscript # amt-2019-53 has received comments from two reviewers, for which you are encouraged to prepare and provide a response along with a revised version of the manuscript.

While reading through the manuscript, I realized that the derived direct radiative effects of aerosols above clouds are correlated with AERONET AOT measured at Ascension Island for 2016 and 2017. Though a general correlation is observed between the two quantities, comparing large area-averaged aerosol effects with AOT from a station might result in mismatch due to sampling differences, i.e., above-cloud versus

cloud-free, difference in observed conditions, i.e., above-cloud versus total column, and so on.

The strength of above-cloud aerosol direct radiative effects is governed by aerosol amounts above cloud as well as brightness (COD) of cloud underneath the aerosol layer. Therefore, it is expected that derived radiative effects should correlate with the above-cloud AOT such as provided by the OMACA product of OMI [Jethva et al., 2016].

The author is strongly recommended to include a similar comparison chart (Figure 7) showing a time-series of area-averaged above-cloud AOT from OMI for 2016 and 2017. Please also retain the AERONET AOT time-series as it is. The OMI/OMACA Level 2 product is freely accessible from the AVDC's data holding portal https://avdc.gsfc.nasa.gov/pub/data/satellite/Aura/OMI/V03/L2/OMACA/. Please contact me back should you face difficulties in locating/reading the OMACA dataset.

A similar product of above-cloud AOT for the Southeastern Atlantic has also been derived using the 'color ratio' information [Jethva et al., 2013] and multi-spectral MODIS observations [Meyer et al., 2015]. Adding the results of these two products to Figure 7 would further confirm the consistency (or lack thereof) between the radiative effects estimated in this paper and satellite products of above-cloud AOT.

To get the access of MODIS multi-spectral ACAOT regional product, please reach out to Kerry Meyer (kerry.meyer@nasa.gov). The 'color ratio' product of ACAOT hasn't been made to the public yet for which the link to download the data will be sent in a separate communication.

We are looking forward to your response and revised paper.

Best,

Hiren Jethva Associate Editor, AMT

---

## Author Comment (AC1) · 28 Jun 2019

*Summary: This paper documents a method called differential aerosol absorption (DAA) to estimate the direct radiative effect (DRE) by the smoke aerosols above cloud (AAC) in the SE Atlantic region using the combination of OMI and MODIS. In this paper, the physical basis of this method is illustrated using selected cases, the uncertainties are analyzed. Applying this method to Aug. 2006 yields an "average aerosol DRE" of 31.5 Wm-2. The topic of this paper is a good match of AMT. The DAA method described in this paper is unique and interesting, although most aspects of this method have already described in early studies i.e., de Graaf et al. (2012, 2014). Overall, I think this paper can eventually be accepted for publication in AMT, but not without significant revision. Below is a list of my major questions and comments. They need to be addressed carefully and thoroughly so the revised paper can meet the standards for publication in AMT.*

The reviewer is thanked for the careful evaluation of the paper. The reviewer raises many questions with respect to the quality of the method and the associated uncertainties. We have tried to address all of the questions of the reviewer as best as possible, and clarified the text to more clearly show the strength and weaknesses of the DAA method. We feel the paper has benefited greatly from the improvements in the text and the added analyses and we thank the reviewer for the feedback.

*Major concerns/questions:*

*1) Uncertainties associated with the anisotropy factor B: Eq. (3) is the main theoretical foundation of the DAA method. I think it needs to be explained better than what is in the current manuscript. A main uncertainty I can see is the anisotropy factor B, which is basically the angular distribution model (ADM) used by the CERES to convert the directional radiance to hemispheric flux. In this study it is assumed that the anisotropy factor for AAC is the same as that for clean clouds. But this assumption is not justified or discussed in depth in the paper. It is simply stated that the uncertainty associated with this factor was investigated in de Graaf et al. (2012). Of course, this is not satisfying and sufficient. This uncertainty needs to be carefully addressed in the present study. In particular, the following questions need to be clarified with proper figures, data and/or references.*

The uncertainty associated with the anisotropy factor has been raised a number of times before by Dr. Zhang in the past. It is not an issue unique to DAA. E.g. CERES measurements also use the assumption of an unchanging anisotropy factor in their forcing computation. The aerosol direct effect is a function of a scene with and without aerosols, and can by definition never be determined by measurements alone, because both scenes do not exist at the same time. Therefore, RTM calculations have to be performed, with assumptions on either cloud properties, aerosol properties or both. The problem here is that (i) cloud properties can be biased by the presence of aerosols and

(ii) aerosol properties by clouds. The DAA method makes markedly different assumptions than methods that derive COT and AOT and compute DRE using RTM, providing independent validation measurements for these methods. Therefore, these assumptions have been described thoroughly in De Graaf et al (2012). The anisotropy factor was addressed as well, for one representative case, and found to be small.

A more complete and extensive study was performed in 2016 by R.E. Prouty, a master student under the supervision of Dr. Zhang. His master thesis work was complemented with SCIAMACHY DRE analyses and described rather completely the uncertainties associated with the anisotropy factor. Unfortunately, this work was never published in the peer-reviewed literature. However, the master thesis is still publicly available (Prouty, 2016). To address the concerns raised by Dr. Zhang, the analyses in Prouty (2016) have been repeated and the main conclusions added to the manuscript in a separate section. The separate questions are answered below.

*a. Anisotropy factor is a strong function of solar-satellite viewing geometry. The uncertainty can be especially large over the special scattering angles, such as rainbow directions. A figure is needed to show the difference between the anisotropy factors for AAC and for clean clouds as a function of satellite viewing direction (i.e., polar contour). This figure can be plotted using the typical solar zenith angle in July or August in the SE Atlantic region at the A-Train crossing time (i.e., 1:30 PM).*

This is correct. Figures have been added following Prouty (2016) to show the change of BRDF of a cloud scene for overlying aerosols. They show that the largest change can be found in the cloud bow (single scattering angles around 140°) for optically thin clouds and (obviously) thick aerosol plumes. The largest change in associated DRE was about 11 Wm$^{-2}$, which is within the error estimate for the OMI/MODIS DRE. However, since the DRE for this case is small, the change due to the anisotropy factor changes the sign of the DRE. Therefore, the assumption on anisotropy factor clearly determines the critical albedo for which the aerosol direct effect changes sign, when estimated using DAA.

*b. Moreover, the anisotropy factor for AAC is also dependent on the scattering properties of the aerosol. It has to be explained why simply assuming the same anisotropy factor B for all types of AAC is sufficient.*

Obviously, this is true for every assumption on aerosol properties. All methods of deriving aerosol DRE assume an aerosol model, mostly fixed based on location, and in the best case varying the SSA. Assuming a wrong aerosol model (e.g. a dust model where smoke is appropriate) may be as disastrous as assuming no aerosol effect at all.

However, in our papers we assume smoke aerosols, and restrict our analysis to the south-east Atlantic during the biomass burning season in Africa, because we show that smoke aerosols have the smallest bias on the retrieved cloud parameters in our method, and smoke also has a small effect on the cloud BRDF. This would be quite different for e.g. dust, and therefore dust is explicitly excluded in the papers.

*c. Similarly, the anisotropy factor for AAC is also dependent special wavelength. The*

*spectral difference between the B for AAC and clean clouds also needs to be addressed and the uncertainty assessed.*

The analysis by Prouty (2016) showed that the largest effect can be expected at UV-vis wavelengths, where the angular effect of aerosol scattering is largest. At SWIR wavelengths the effect of aerosols is much smaller and more smooth, largely canceling the BRDF change. In the revised manuscript the effects at 555 nm and 2130 nm are compared.

*2) The difference between OMI and MODIS cloud reflectance: On page 9 line 20, it is found that "Clearly there is a mismatch between OMI and MODIS for the broken cloud scene, which is caused by rapid cloud changes. The averaged reflectance of the scene has changed during the 15 minutes between overpasses Aura and Aqua." First, I found the difference between the OMI and MODIS cloud reflectance surprisingly large (i.e., 0.6 MODIS vs. 0.4 OMI). So this seems to be a Second, I found the speculation that this difference is caused by "rapid cloud changes" not convincing. Note that the underlying clouds below AAC in SE Atlantic region are mostly boundary stratocumulus clouds. These clouds are pretty stable. It is hard to imagine cloud reflectance changes 50% in only 15 minutes. Min and Zhang (2014) studied the influence of cloud diurnal cycle on the DRE estimation on the basis of MODIS and SEVIRI observations. They found about 5% cloud fraction change in the SE Atlantic region between Terra (10:30 AM) and Aqua (1:30 AM). The two satellites are separated by 3 hours and the cloud fraction change is only 5%. I am not convinced that within 15 minutes the cloud reflectance can change 50% (for convective clouds maybe, but not for stratocumulus).*

*So this issue/question needs to be addressed and clarified with substantial evidences. I'd suggest the author to use the high-frequency SEVIRI data (15 minutes) to assess how much the cloud property changes within 15 minutes in the studied region.*

This question is slightly surprising. Surely, the reflectance in an OMI pixel can change substantially in 15 minutes. The change in reflectance in a pixel due to cloud contamination depends on the wind speed and the size of a pixel. An OMI nadir pixel is 13x24 km$^2$. To completely fill a cloud-free pixel in 15 minutes, the clouds only have to move in the along-track direction at a speed of $13 \times 4 = 52$ km/h. A cloud fraction of 0.2 is already more than enough to change the reflectance by more than 50% over the dark ocean background, so a mere 10 km/h would suffice. The presented change of 0.4 to 0.6 should not be surprising.

The reviewer probably mixes average values with individual ones. Min and Zhang (2014) present analyses of the cloud heterogeneity, using histograms which are based on a large number of MODIS/Aqua and MODIS/Terra pixels, with many compensating effects. Even then, CF change is significant. Min and Zhang (2014) conclude that marine boundary layer clouds have significant small-scale heterogeneity. However, these numbers are quite different from individual cases, for which the reflectance can change easily, as explained above. Our example of one OMI pixel in which the reflectance change was large due to moving broken cloud fields merely illustrates our strategy of combining OMI and MODIS reflectances, for measurements that are 15 minutes apart. This is relevant for understanding the method described in the manuscript. Differences

Cloud Fraction in Retrieval Region (5x5 1-km Pixels) from 1-km Cloud Mask

[Figure]

Figure 1: MODIS cloud fraction on 1 August 2006 13:14:09 and 13:14:15 UTC. The OMI pixels were acquired at 13:30:15 and 13:30:21 UTC, respectively

between the OMI and MODIS reflectances occur often, but our strategy to combine them works very well for the derivation of DRE, as explained in the manuscript. The issue was also addressed in (de Graaf et al., 2016).

The OMI FRESCO effective cloud fractions for the pixels in Fig.2 in the manuscript were given in the panels with the spectra. They were 0.69 and 0.35, respectively. Effective cloud fractions are generally smaller than geometric cloud fractions. I tried to determine the MODIS geometric cloud fractions of the pixels using the L2 data cloud data from the MODIS MYD06 dataset. Fig 1 shows the L2 5x5 km2 cloud fraction from 1x1 data in the same area as in Fig.2b of the manuscript, at the time of MODIS overpass, which is 15 minutes before OMI. It shows the open cloud fields just at the edge of the lowest OMI pixel. The most common wind in this area at the surface is from the southeast, and this would have moved the cloud edge over the blue OMI pixel, lowering the FRESCO eff. CF for this pixel to 0.35.

A better way of determining the geometric cloud fraction in the OMI footprint would be to count MODIS pixels with cloud mask on and off, but this was not further attempted.

"Rapid cloud changes" is a misleading term though. It should be "significant reflectance changes due to changes in cloud fraction". We have removed the term from the manuscript, and rephrased the sentence more carefully.

*3) Sampling rate of the DRE needs to be provided: In my opinion, the DRE values are only meaningful and useful when the corresponding sampling rate is given side by side. It seems to me that the DAA method described here is only applicable to certain portion of the total cloud fraction. But the paper provides no data or analysis of the sampling rate. As shown in Zhang et al. (2016) as well as many previous studies, the DRE is dependent on both AOT and COT (See their Figure 9a). If a method only samples, say large COT and large AOT, then the DRE from such method would yield larger DRE than another method that can sample all COT and AOT. But the results from the two methods are not directly comparable. Because of the lack of sampling rate information, it will be difficult for other researchers to compare or use the DRE results from this study.*

Indeed, polar orbiting satellites only sample the atmosphere at one particular time. And DAA is only applicable to a certain portion of the total cloud fraction, i.e. for scenes that are sufficiently cloudy. Therefore, the title states explicitly that only cloud scenes are considered, and the DRE will be (mostly) positive. Furthermore, in this section 4.2 and Figs 5 and 6 is was explicitly stated that only scenes with a CF > 0.3 were selected. However, the statements were absent from the conclusions and in the abstract, and the explicit CF sampling has been added there as well.

*To address this issue, I believe the following sampling rates need to be provided with proper figures or tables:*
*a. What is the total cloud fraction identified by the collocated OMI-MODIS observations?*

A minimum of CF=0.3 is always adopted, as stated in section 4.2 and Figs 5 and 6. In Fig 2. it was shown that the CF for the two scenes were quite different, 0.69 and 0.35 respectively, as was indicated in the figures.

*b. What is the fraction of cloud with detectable AAC, e.g., UV-AI >2?*

There is no filter on any aerosol or reflectance conditions. All cloud scenes with CF>0.3 and CP<800 hPa were processed, see section 4.2. Scenes without aerosols will yield zero DRE (ideally, see Fig. 6).

*c. What is the fraction of the cloud with valid DRE estimation using the DAA method?*

Valid cloud retrievals are possible for scenes with a minimum CF of about 0.15. The exact number was not analysed. This is, however, irrelevant, since only scenes with a CF >0.3 are considered. Cloud information is included in the dataset, though.

*d. The above information should be provided for all DRE results, for example, Figure 5. I am wondering to what extent the inter-annual difference in Figure 5 is due to real cloud or aerosol and to what extent it is actually due to year-to-year sampling rate difference.*

All of it, since SCIAMACHY and OMI/MODIS DRE were compared for only those scene that had a CF > 0.3. As was explicitly stated in the text, and the caption of Fig.5.

*e. Another relevant question is: What are the DREs for the cloud with detectable AAC but the DAA method cannot be used for any reasons?*

Obviously, it is difficult to present the DRE using DAA for those scenes that the DAA fails.

The cloud retrievals fail for CF around 0.15 or smaller. Scenes with 0.15 < CF < 0.3 are filtered for the analyses. The aerosol DRE for cloud-free scenes (down from CF < 0.3) can and have been analyzed with different techniques than DAA, e.g. Chand et al. (2009); Jethva et al. (2013); Meyer et al. (2015). The current study is not suitable nor intended to answer this question.

*The sampling rates need to be provided whenever the DRE values are given, i.e., in the abstract and conclusion.*

Indeed, the sampling rates were not repeated in the abstract and conclusions. This omission has been corrected.

*4) The DRE results need to be presented more carefully: It needs to be empathized in the abstract and conclusion that the DRE from this study is the instantaneous DRE only at the A-Train over passing time. When talking about the "daily averaged" DRE, the following questions need to be clarified: a. Is it a diurnal average, i.e., including nighttime, or only daytime average? See Zhang et al. (2016) about diurnal average. b. Does the daily average account for the diurnal cycle cloud clouds? See Min and Zhang (2014) about the impact for cloud diurnal cycle on DRE estimation.*

This has been corrected. Indeed, the average values were area-averaged only, for each day. "Daily area averaged values" is ambiguous, "Daily, area-averaged values" was intended. However, the term has been dropped entirely, to unambiguously state that "area-averaged instantaneous DRE values" are presented for each day. This has been changed throughout the manuscript.

*Reviewer #2*
*This paper describes a method to estimate the direct radiative effect (DRE) of aerosols above clouds using OMI and MODIS measurements. The technique (differential aerosol absorption, DAA) is somewhat different from related algorithms as is in essence a retrieval of the radiative effect itself rather than being focused on the optical/microphysical quantities of the aerosols and clouds, which gives it a different set of strengths and weaknesses from other above-cloud aerosol algorithms. The DAA retrieval is an extension of an algorithm published by the authors previously, which used SCIAMACHY instead. The SCIAMACHY record ended in 2012, while OMI/MODIS are still flying, and other sensors with similar capabilities also fly now and are planned for the future (e.g. OMPS/VIIRS, and PACE OCI). SCIAMACHY was a spectrometer with a coarse footprint, while OMI is a UV-vis spectroradiometer and MODIS is multispectral. MODIS and OMI also have different footprints (both finer than SCIAMACHY) and fly on different platforms. So, the adaptation of the algorithm from SCIAMACHY to these other sensors is of scientific interest and sufficiently non-trivial and novel. This work is well in scope for AMT.*
*The quality of language is good. With the exception of Section 5, which was a let-down, the paper is pretty good. Overall I recommend publication after minor revisions; some points in the text need expanding and I have a few concerns with the error budget, as well as the lack of use of ORACLES data. I would be happy to review the revision.*

**Comments/Corrections** *Page 2, lines 23-27: While POLDER is probably the most informative, there are several techniques to estimate above-cloud AOT and COT from MODIS and/or OMI alone. See e.g. work by Meyer, Sayer, Jethva for various algorithms. I'm not saying that the authors have to cite each paper in this field, but a brief acknowledgment/discussion of the fact that there are several MODIS or OMI techniques which have been developed and used successfully already, and it's not only POLDER and CALIOP which have these capabilities, would be welcome.*

This is very correct observation. All the references to other methods ended up in the accompanying paper about the OMI/MODIS - POLDER comparison. The references have been added to this manuscript as well, it was no intention to disregard the work done by other authors.

*Page 2, line 26: This mentions that a comparison with POLDER results is presented elsewhere. I went to the bibliography and this is listed as a study in preparation for submission to GRL. If this work has already been done, it would be good to briefly summarize the results. This is relevant because the POLDER technique is quite different from DAA. Otherwise, I'd just say that the comparison will be performed and remove the citation. I suppose the progress of both papers can be assessed at the time that this manuscript is revised. Given this paper is cited again on page 14, I think it's important that we get to see the results, which we can't because the paper being cited*

*hasn't even been submitted yet. Basically, either give us the information or remove the citation.*

The information has been added and the citation removed.

*Page 3, line 5: I'd add a brief discussion of and references in support of the assumption of negligible aerosol effects in the longwave. While agree it is probably the case for smoke, it may not be for dust. I know there are various papers looking at shortwave vs. radiative effects of dust under various conditions (e.g. over land, ocean, daytime, nightime, cloud). I think it's important to acknowledge when/where this assumption is reasonable and the magnitude of the error from assuming it is negligible. Some readers might otherwise assume it is always negligible. This is mentioned later on page 5, but I'd state it here too.*

Correct, the application to dust aerosols will fail. The restriction to smoke aerosols was added.

*Equation 3: this is the core of the method; the most questionable assumption here seems to me to be that the anisotropy factor B is the same for an aerosol-laden and an aerosol-free cloud. Intuitively one would expect the aerosol-laden scene to be less anisotropic. Page 4 directs the reader to de Graaf (2012), and I found that their Section 6.2 addresses this. I realise that these errors are often AOT-dependent but to give the reader a rough idea of expected performance for the SCIAMACHY case (as a reference for the present MODIS/OMI), I suggest summarizing this information here (either the total figure of 8 Wm-2 given in section 6.3 of the 2012 paper, or a brief quantification of the individual components) so the reader does not need to dig out the previous paper.*

This section has been extended with a thorough analysis of the uncertainty due to anisotropy factor, following the suggestion of reviewer #1. In addition, the reference to SCIAMACHY results have been removed, and an error estimate for OMI/MODIS measurements only has been added.

*Section 4.3: If I understand this correctly, the biggest contribution to the retrieval error is estimated as the calculated forcing for pixels where the UV aerosol index (UVAI) is less than 0. This has mean and standard deviation 7 and 12 Wm-2 respectively. This is fine in theory but I have some questions in practice. UVAI is a semi-quantitative detection since it depends on not only aerosol absorption but also on factors including solar/view geometry, altitude, cloud properties, underlying surface (in cases of broken cloud) etc. The threshold value of 0 is not supported by radiative transfer arguments as far as I can tell, but rather seems a hand-waving threshold that is a nice round number. While sensible as a first approximation it is certainly possible to get negative UVAI when there is some absorption (this is even shown in the de Graaf 2005 paper the authors cite at this point), or positive when the aerosols are only scattering; while one might argue that this would contribute to the scatter in Figure 6, there is no reason to assume that it would lead to an unbiased estimate. Thus the reported systematic bias of 7 W m-2 might be true, or might be the result of choosing*

*0 as the UVAI threshold when another one would be more appropriate. It is not clear which UVAI the authors are using (there are several definitions and data versions). I believe the latest OMI standard product version includes a new definition and calculation which reduces the dependence on factors like geometry (see Torres et al 2018, https://doi.org/10.5194/amt-11-2701-2018 ). If this was not what was used, I recommend repeating this analysis with it. The new OMI UVAI will reduce some of these confounding effects such that it is a better proxy for aerosol absorption. It should make the authors' assessment of systematic/random errors here more realistic. So, my suggestions are: (1) Ensure that the latest OMI UVAI data set is used for this calculation, to decrease the confounding non-aerosol effects, and (2) acknowledge that UVAI=0 as a threshold is arbitrary and mention (or even better), estimate the additional uncertainty this is contributing to the error analysis in section 4.3. Perhaps a better threshold than UVAI=0 could be determined and adopted.*

This is a very much appreciated observation by the reviewer. The used verion of the AI was the OMAERO AI v1.2.3.1 developed and maintained at KNMI, using the 354/388 nm wavelength pair. A small analysis on the dependence on the threshold showed a decrease in bias with decreasing AI threshold, see the table below. The average DRE is 1–2 for UVAI down to -1.5 and -1.0, below which too few pixels remain. This is lower than the mean of 7 that we found for the indeed arbitrary threshold of 0.0. Increasing the threshold further increases the mean, as expected, as more and more pixels with absorbing aerosols are incorporated. So, indeed the assumed bias seems

Table 1: UVAI threshold analysis results

OMAERO v 1.1.1

| AI | bias | std. dev. | number of scenes |
|---|---|---|---|
| -1.5 | 2 | 10 | 44 |
| -1.0 | 1 | 12 | 368 |
| -0.5 | 4 | 11 | 2740 |
| 0 | 7 | 12 | 12471 |
| 0.5 | 10 | 13 | 32067 |
| 1.0 | 13 | 15 | 59564 |
| 1.5 | 15 | 16 | 89021 |
| 2.0 | 17 | 17 | 109480 |

OMAERUV 1.8.9.1 (2017)

| UVAI | bias | std. dev. | number of scenes |
|---|---|---|---|
| -1.5 | 17 | 15 | 9 |
| -1.0 | 12 | 13 | 87 |
| -0.5 | 13 | 11 | 899 |
| 0 | 15 | 12 | 10579 |
| 0.5 | 18 | 12 | 46454 |
| 1.0 | 21 | 13 | 103513 |
| 1.5 | 24 | 14 | 152008 |
| 2.0 | 24 | 14 | 180830 |

[Figure]

Figure 2: Scatterplot of OMAERUV UVAI v1.8.9.1 vs OMAERO AI v1.2.3.1

to disappear with more stringent filtering on AI. Interestingly, the standard deviation does not change much with AI threshold, suggesting that the standard deviation is a good estimate of the random error, i.e. the ability to correctly simulate a cloud scene spectrum and estimate the DRE from that.

The analysis was repeated with the new OMAERUV UV-AI developed at NASA. The definition for this aerosol index is very different than the OMAERO AI. The influences from cloud scattering is included in the index using simple scattering layers in the RTM-generated LUTs, using Mie or HG clouds. The OMAERO AI and OMAERUV UV-AI were compared in Fig. 2 for all 2006 scenes in this study (with and without aerosols). As the figure shows, their is a strong correlation between the products, but there are also very clear differences.

A repetition of the analysis above showed that the OMAERUV UVAI seems unsuitable for removing absorbing aerosols in cloud scenes, see table 1. With different UVAI thresholds the average DRE is always significantly higher than 0. The reason is unclear, but maybe for fully clouded scenes the effects of simulating cloud reflectances in the LUTs is so large that the aerosol effects are not significant anymore.

*Section 5: Honestly this section is a bit of a let down and missed opportunity. The authors show time series of radiative effect during the CLARIFY, ORACLES, and LA-SIC campaigns, and give citations about them. However the analysis amounts to plotting back-trajectories and showing tmie series of AERONET AOT against DRE. None of the actual data from the field campaigns appears to have been actually used. The ORA-CLES data are already freely available from https://espoarchive.nasa.gov/archive/browse/oracles. This includes a large number of relevant observations including e.g. irradiance/flux which could be used to evaluate the algorithm's output more quantitatively, rather than*

*just showing that AOT at Ascension Island is correlated with DRE over the southern Atlantic Ocean. I strongly urge the authors to look at these data as there are bound to be some matches close in space/time to the A-Train overpass. It would help give a sense of whether the DRE magnitudes are reasonable, as right now all we can say is that temporal variation seems reasonable. As-is, the paper's introduction and section 5 state these plots are presented "in support of" these campaigns, but there's really no linkage demonstrated in what's actually contained in the paper.*

We fully agree with the reviewer, and a comparison with aircraft would be very valuable. We have tried to add comparisons with ORACLES data, which are indeed freely available. We also contacted individual researchers in the ORACLES community. Unfortunately, it was not possible to add anything significant within the time frame of the manuscript review period. The analysis of aircraft data is specialized work, and a thorough comparison deserve more time than was available here. A separate publication would be more suitable for this.

The suggestion of the editor was followed to compare with satellite AOT from OMI and MODIS, to at least present some more evidence of correctness of the DRE magnitude. Also, all references to the support of the aircraft campaigns were removed. The manuscript now states the existence of the campaigns and the data, and merely illustrates the DRE data during this period, as it was intended.

*Conclusions: this quotes mean and standard deviations of DRE. I'd be interested to see some pdfs somewhere in the paper, to see what the distributions look like at different scales. If they are skewed then mean and standard deviation might not be the best summary metrics, perhaps median and interquartile range would be better. This could also be something to add to the SCIAMACHY comparison section, for example: show whether the pdfs of DRE are similar to within the expected level of consistency for e.g. a season's worth of data over the south Atlantic. This would complement the existing instantaneous consistency assessment with a more climatological consistency assessment, which is after all important if the end goal is to move toward a long-term post-SCIAMACHY record.*

A figure of histograms of OMI/MODIS DRE and SCIAMACHY DRE has been added.

*I was also surprised not to see any mention of VIIRS/OMPS in the paper. These sensors fly on SNPP (since 2011– there's even a brief overlap with the SCIAMACHY record) and NOAA20 (since 2017), and have similar capabilities overall to MODIS/OMI. In some senses they would even be a better choice than the MODIS/OMI pair, because they fly on the same satellite, which simplifies some of the collocation/time difference issues. Again, I don't expect the authors to demonstrate the algorithm with VIIRS/OMPS, but a brief mention that this sensor combination exists and the relative merits of the sensor pair would be welcome.*

True. This has been added to the conclusions.
We thank the reviewers for the helpful comments to improve the manuscript.

**References**

Chand, D., Wood, R., Anderson, T. L., Satheesh, S. K., and Charlson, R. J.: Satellite-derived direct radiative effect of aerosols dependent on cloud cover, Nat. Geosci., 2, https://doi.org/10.1038/NGEO437, 2009.

de Graaf, M., Sihler, H., Tilstra, L. G., and Stammes, P.: How big is an OMI pixel?, Atmos. Meas. Tech., https://doi.org/10.5194/amt-9-3607-2016, URL `http://www.atmos-meas-tech.net/9/3607/2016/`, 2016.

Jethva, H., Torres, O., Remer, L. A., and Bhartia, P. K.: A Color Ratio Method for Simultaneous Retrieval of Aerosol and Cloud Optical Thickness of Above-Cloud Absorbing Aerosols From Passive Sensors: Application to MODIS Measurements, IEEE T. Geosci. Remote, 51, 3862–3870, https://doi.org/10.1109/TGRS.2012.2230008, 2013.

Meyer, K., Platnick, S., and Zhang, Z.: Simultaneously inferring above-cloud absorbing aerosol optical thickness and underlying liquid phase cloud optical and microphysical properties using MODIS, J. Geophys. Res., 120, 5524–5547, https://doi.org/10.1002/2015JD023128, 2015.

Prouty, Jr., R. E.: Impact of above-cloud aerosols on the angular distribution pattern of cloud bidirectional-reflectance and implication for above-cloud aerosol direct radiative effect, MSc. thesis ISBN: 9781369654653, University of Maryland, 2016.

---

## Author Comment (AC2) · 28 Jun 2019

All the issues raised by the reviewers have been addressed in the supplement 'amt-2019- 53-supplement.pdf'. The manuscript has been revised to reflect the answers to the reviews. Furthermore, the suggestions in the interactive comment by the editor was also implemented.
* * *

---

## Referee Report (RR1)

Review on "*Aerosol direct radiative effect over clouds from a synergy of OMI and MODIS reflectances*" by de Graaf et al.

I had three major concerns/questions for the original manuscript. The first is about the anisotropy factor, the second is about why OMI and MODIS observed cloud reflectances differ significantly when their overpassing time is only 15 minutes away, and the last question is about the sampling rate of method described in this paper for deriving DRE of above cloud smoke. The authors have addressed these major concerns/questions carefully and thoroughly.

However, I still have a few minor questions and comments left. They have to be addressed before the manuscript can be accepted for publication.

- Even in the revised manuscript, the definition of the DRE derived from the combined OMI-MODIS observation is still not clear and precise enough. As pointed out in *[Zhang et al.*, 2016], the all-sky DRE of aerosol is defined as $DRE_{all-sky} = f_c\overline{DRE_{cloudy}} + (1 - f_c)\overline{DRE_{clear}}$, where $f_c$ is the cloud fraction, $\overline{DRE_{cloudy}}$ and $\overline{DRE_{clear}}$ is the averaged cloudy-sky and clear-sky DRE, respectively. Take a hypothetical example. Assuming that we have an OMI-MODIS pixel with a cloud fraction $f_c = 0.5$. The $\overline{DRE_{cloudy}}$ due to above-cloud smoke is 40 Wm$^{-2}$ and $\overline{DRE_{clear}}$ is 1 Wm$^{-2}$. Which of the following values does the method described in this paper reports? 1) $\overline{DRE_{cloudy}}$=40 Wm$^{-2}$, 2) $f_c\overline{DRE_{cloudy}}$=0.5*40 Wm$^{-2}$=20 Wm$^{-2}$, or 3) $DRE_{all-sky}$=20 Wm$^{-2}$ +0.5*1 Wm$^{-2}$=20.5 Wm$^{-2}$. This question should be clarified early in the paper, for example, in Section 2. It is an important question because the answers will help the readers understand precisely the meaning of the DRE from this study, as well as how to compare the DRE from this study with previous ones such as *[Zhang et al.*, 2016].
- Another question, which is related to the question above, is about how to scale the OMI spectrum to match MODIS observation. If I understand correctly, the reflectance of a cloudy pixel observed by OMI can be decomposed into $R_{OMI} = f_{c,OMI}R_{cld+aer} + (1 - f_{c,OMI})R_{clr}$. Similarly, the reflectance observed by MODIS is $R_{MODIS} =$

$f_{c,MODIS}R_{cld+aer} + (1 - f_{c,MODIS})R_{clr}$. It is not clear to me what the "scaling" in section 3.5 means. Is the "scaling" intended to match $R_{OMI}$ and $R_{MODIS}$? What is the "scaling" factor and what is its physical meaning? These questions are important, and they need to be clarified in the context of the above equations.

- Page 2 line 20, there are a few noteworthy previous studies on the DRE of above cloud aerosols that might deserve being cited here, e.g., *[Peters et al.*, 2011; *Feng and Christopher*, 2015; *Zhang et al.*, 2016] *and a very recent study [Kacenelenbogen et al.*, 2019]. Some discussion should be made about the originality and significance of the current study w.r.t. these previous studies as well as those from the leading author.

- Page 4, equation (3), again what is the exact definition of $DRE_{aer}$ here? See my first and second questions above.

- Page 7, similarly, what is the DRE derived from *SCIAMACHY?* Is it $\overline{DRE_{cloudy}}$, $f_c\overline{DRE_{cloudy}}$ or $DRE_{all-sky}$?

- Page 10, line3, "*and 0.35 in the red pixel*". Should it be "*and 0.35 in the blue pixel*"

- Also, what does FRESCO stand for?

*Feng, N., and S. A. Christopher (2015), Measurement-based estimates of direct radiative effects of absorbing aerosols above clouds, Journal of Geophysical Research-Atmospheres*, *120*(14), 2015JD023252–n/a, doi:10.1002/2015JD023252.

Kacenelenbogen, M. S. et al. (2019), Estimations of global shortwave direct aerosol radiative effects above opaque water clouds using a combination of A-Train satellite sensors, *Atmospheric Chemistry and Physics*, *19*(7), 4933–4962, doi:10.5194/acp-19-4933-2019.

Peters, K., J. Quaas, and N. Bellouin (2011), Effects of absorbing aerosols in cloudy skies: a satellite study over the Atlantic Ocean, *Atmos. Chem. Phys*, *11*, 1393–1404.

Zhang, Z., K. Meyer, H. Yu, S. Platnick, P. Colarco, Z. Liu, and L. Oreopoulos (2016), Shortwave direct radiative effects of above-cloud aerosols over global oceans derived from 8 years of CALIOP and MODIS observations, *Atmospheric Chemistry and Physics*, *16*(5), 2877–2900, doi:10.5194/acp-16-2877-2016.

---

## Editor Decision (ED1)

Reviewer # 1

Review on "*Aerosol direct radiative effect over clouds from a synergy of OMI and MODIS reflectances*" by de Graaf et al.

I had three major concerns/questions for the original manuscript. The first is about the anisotropy factor, the second is about why OMI and MODIS observed cloud reflectances differ significantly when their overpassing time is only 15 minutes away, and the last question is about the sampling rate of method described in this paper for deriving DRE of above cloud smoke. The authors have addressed these major concerns/questions carefully and thoroughly.

However, I still have a few minor questions and comments left. They have to be addressed before the manuscript can be accepted for publication.

- Even in the revised manuscript, the definition of the DRE derived from the combined OMI-MODIS observation is still not clear and precise enough. As pointed out in *[Zhang et al.,* 2016], the all-sky DRE of aerosol is defined as $DRE_{all-sky} = f_c\overline{DRE_{cloudy}} + (1 - f_c)\overline{DRE_{clear}}$, where $f_c$ is the cloud fraction, $\overline{DRE_{cloudy}}$ and $\overline{DRE_{clear}}$ is the averaged cloudy-sky and clear-sky DRE, respectively. Take a hypothetical example. Assuming that we have an OMI-MODIS pixel with a cloud fraction $f_c = 0.5$. The $\overline{DRE_{cloudy}}$ due to above-cloud smoke is 40 Wm$^{-2}$ and $\overline{DRE_{clear}}$ is 1 Wm$^{-2}$. Which of the following values does the method described in this paper reports? 1) $\overline{DRE_{cloudy}}$=40 Wm$^{-2}$, 2) $f_c\overline{DRE_{cloudy}}$=0.5*40 Wm$^{-2}$=20 Wm$^{-2}$, or 3) $DRE_{all-sky}$=20 Wm$^{-2}$ +0.5*1 Wm$^{-2}$=20.5 Wm$^{-2}$. This question should be clarified early in the paper, for example, in Section 2. It is an important question because the answers will help the readers understand precisely the meaning of the DRE from this study, as well as how to compare the DRE from this study with previous ones such as *[Zhang et al.,* 2016].
- Another question, which is related to the question above, is about how to scale the OMI spectrum to match MODIS observation. If I understand correctly, the reflectance of a cloudy pixel observed by OMI can be decomposed into $R_{OMI} = f_{c,OMI}R_{cld+aer} + (1 - f_{c,OMI})R_{clr}$. Similarly, the reflectance observed by MODIS is $R_{MODIS} =$

$f_{c,MODIS}R_{cld+aer} + (1 - f_{c,MODIS})R_{clr}$. It is not clear to me what the "scaling" in section 3.5 means. Is the "scaling" intended to match $R_{OMI}$ and $R_{MODIS}$? What is the "scaling" factor and what is its physical meaning? These questions are important, and they need to be clarified in the context of the above equations.

- Page 2 line 20, there are a few noteworthy previous studies on the DRE of above cloud aerosols that might deserve being cited here, e.g., *[Peters et al., 2011; Feng and Christopher, 2015; Zhang et al., 2016] and a very recent study [Kacenelenbogen et al., 2019]*. Some discussion should be made about the originality and significance of the current study w.r.t. these previous studies as well as those from the leading author.

- Page 4, equation (3), again what is the exact definition of $DRE_{aer}$ here? See my first and second questions above.

- Page 7, similarly, what is the DRE derived from *SCIAMACHY*? Is it $\overline{DRE_{cloudy}}$, $f_c\overline{DRE_{cloudy}}$ or $DRE_{all-sky}$?

- Page 10, line3, "*and 0.35 in the red pixel*". Should it be "*and 0.35 in the blue pixel*"

- Also, what does FRESCO stand for?

*Feng, N., and S. A. Christopher (2015), Measurement-based estimates of direct radiative effects of absorbing aerosols above clouds, Journal of Geophysical Research-Atmospheres, 120*(14), 2015JD023252–n/a, doi:10.1002/2015JD023252.

Kacenelenbogen, M. S. et al. (2019), Estimations of global shortwave direct aerosol radiative effects above opaque water clouds using a combination of A-Train satellite sensors, *Atmospheric Chemistry and Physics, 19*(7), 4933–4962, doi:10.5194/acp-19-4933-2019.

Peters, K., J. Quaas, and N. Bellouin (2011), Effects of absorbing aerosols in cloudy skies: a satellite study over the Atlantic Ocean, *Atmos. Chem. Phys, 11*, 1393–1404.

Zhang, Z., K. Meyer, H. Yu, S. Platnick, P. Colarco, Z. Liu, and L. Oreopoulos (2016), Shortwave direct radiative effects of above-cloud aerosols over global oceans derived from 8 years of CALIOP and MODIS observations, *Atmospheric Chemistry and Physics, 16*(5), 2877–2900, doi:10.5194/acp-16-2877-2016.

Reviewer # 2

I reviewed a previous version of this manuscript. My main issues with the previous version were (1) discussion of the anisotropy factor B and related uncertainty; (2) choice of a threshold UVAI=0 as a baseline for the uncertainty calculations; and (3) the fairly simplistic nature of the 2016/2017 data analysis (mentioning ORACLES but not using the data). In this revision the authors have expanded the discussion of (1) and (2), which I appreciate, and added MODIS and OMI above-cloud satellite time series for (3) while noting that the comparison with ORACLES data is better suited for a separate paper (which I hope they do). This makes it more convincing, in my view, than the previous submission.

As a result I do not have technical objections to the publication of this manuscript, although the other reviewer (Z. Zhang) is more of an expert in the forcing aspect than I am, so I would defer to their judgment.

I have a few minor comments, but otherwise find the manuscript acceptable for publication after technical corrections. I would be happy to review these corrections if the Editor feels it would be helpful, although I do not think it is necessary, provided the other reviewer is satisfied.

Previous comment on POLDER: the authors had cited a paper in preparation which compared the OMI/MODIS results against POLDER. I'd suggested the authors provide the results here or remove the reference, since we can't see the results otherwise (given it's a paper which has not been submitted yet). They replied that they have added the information and also removed the citation. It's not clear to me where the information about this comparison has been added, as it doesn't seem to be in the original section; there is a brief mention in section 4.2 of POLDER but that seems to be it? Mentioning just in case something was inadvertently omitted, but I think it is ok as-is.

Page 7 line 27: a reference for the MODIS sensor should be added (one is already provided for the other satellite instruments used). I don't have a particular strong feeling about which, but Salmonson et al (1989) is often used: https://ieeexplore.ieee.org/document/20292

Page 10 line 2: Acronym FRESCO needs to be defined at first use.

Page 21 line 4: Acronym AERONET needs to be defined at first use. Also, state which version you are using and provide a reference. It should be the current version 3, with citation Giles et al (2019): https://www.atmos-meas-tech.net/12/169/2019/amt-12-169-2019-discussion.html Also define the data level being used. I see this is level 1.5 rather than the standard level 2; after checking the AERONET website I see level 2 is not available yet for Ascension Island in 2017. It is worth mentioning the difference between levels and stating why level 1.5 is used here.

Page 22 lines 30-32: VIIRS and OMPS acronyms should be defined at first use. Also, there is more than just SNPP now, NOAA20 (formerly JPSS1) launched in late 2017.

Section 4.3.3 and more generally: in this section (and elsewhere) the authors say "error" often. I think a lot of these times, they really mean "uncertainty". For example, page 19 line 9 I think the authors mean the "uncertainty" in the DRE retrievals, not the error, since we don't have a truth to compare to. If possible it would also be good for the authors to clarify whether the estimates they provide in the paper refer to typical levels of uncertainty (e.g. 1-sigma), maximum likely uncertainty, or similar.

---

## Author Response (AR2)

Reviewer # 1

Review on "*Aerosol direct radiative effect over clouds from a synergy of OMI and MODIS reflectances*" by de Graaf et al.

I had three major concerns/questions for the original manuscript. The first is about the anisotropy factor, the second is about why OMI and MODIS observed cloud reflectances differ significantly when their overpassing time is only 15 minutes away, and the last question is about the sampling rate of method described in this paper for deriving DRE of above cloud smoke. The authors have addressed these major concerns/questions carefully and thoroughly.

However, I still have a few minor questions and comments left. They have to be addressed before the manuscript can be accepted for publication.

- Even in the revised manuscript, the definition of the DRE derived from the combined OMI-MODIS observation is still not clear and precise enough. As pointed out in *[Zhang et al., 2016]*, the all-sky DRE of aerosol is defined as $DRE_{all-sky} = f_c\overline{DRE_{cloudy}} + (1 - f_c)\overline{DRE_{clear}}$, where $f_c$ is the cloud fraction, $\overline{DRE_{cloudy}}$ and $\overline{DRE_{clear}}$ is the averaged cloudy-sky and clear-sky DRE, respectively. Take a hypothetical example. Assuming that we have an OMI-MODIS pixel with a cloud fraction $f_c = 0.5$. The $\overline{DRE_{cloudy}}$ due to above-cloud smoke is 40 Wm$^{-2}$ and $\overline{DRE_{clear}}$ is 1 Wm$^{-2}$. Which of the following values does the method described in this paper reports? 1) $\overline{DRE_{cloudy}}$=40 Wm$^{-2}$, 2) $f_c\overline{DRE_{cloudy}}$=0.5*40 Wm$^{-2}$=20 Wm$^{-2}$, or 3) $DRE_{all-sky}$=20 Wm$^{-2}$ +0.5*1 Wm$^{-2}$=20.5 Wm$^{-2}$. This question should be clarified early in the paper, for example, in Section 2. It is an important question because the answers will help the readers understand precisely the meaning of the DRE from this study, as well as how to compare the DRE from this study with previous ones such as *[Zhang et al., 2016]*.
- Another question, which is related to the question above, is about how to scale the OMI spectrum to match MODIS observation. If I understand correctly, the reflectance of a cloudy pixel observed by OMI can be decomposed into $R_{OMI} = f_{c,OMI}R_{cld+aer} + (1 - f_{c,OMI})R_{clr}$. Similarly, the reflectance observed by MODIS is $R_{MODIS} =$

$f_{c,MODIS}R_{cld+aer} + (1 - f_{c,MODIS})R_{clr}$. It is not clear to me what the "scaling" in section 3.5 means. Is the "scaling" intended to match $R_{OMI}$ and $R_{MODIS}$? What is the "scaling" factor and what is its physical meaning? These questions are important, and they need to be clarified in the context of the above equations.

- Page 2 line 20, there are a few noteworthy previous studies on the DRE of above cloud aerosols that might deserve being cited here, e.g., *[Peters et al.*, 2011; *Feng and Christopher*, 2015; *Zhang et al.*, 2016] *and a very recent study [Kacenelenbogen et al.*, 2019]. Some discussion should be made about the originality and significance of the current study w.r.t. these previous studies as well as those from the leading author.

- Page 4, equation (3), again what is the exact definition of $DRE_{aer}$ here? See my first and second questions above.

- Page 7, similarly, what is the DRE derived from *SCIAMACHY?* Is it $\overline{DRE_{cloudy}}$, $f_c\overline{DRE_{cloudy}}$ or $DRE_{all-sky}$?

- Page 10, line3, "*and 0.35 in the red pixel*". Should it be "*and 0.35 in the blue pixel*"

- Also, what does FRESCO stand for?

*Feng, N., and S. A. Christopher (2015), Measurement-based estimates of direct radiative effects of absorbing aerosols above clouds, Journal of Geophysical Research-Atmospheres, 120(14), 2015JD023252–n/a, doi:10.1002/2015JD023252.*

Kacenelenbogen, M. S. et al. (2019), Estimations of global shortwave direct aerosol radiative effects above opaque water clouds using a combination of A-Train satellite sensors, *Atmospheric Chemistry and Physics*, *19*(7), 4933–4962, doi:10.5194/acp-19-4933-2019.

Peters, K., J. Quaas, and N. Bellouin (2011), Effects of absorbing aerosols in cloudy skies: a satellite study over the Atlantic Ocean, *Atmos. Chem. Phys*, *11*, 1393–1404.

Zhang, Z., K. Meyer, H. Yu, S. Platnick, P. Colarco, Z. Liu, and L. Oreopoulos (2016), Shortwave direct radiative effects of above-cloud aerosols over global oceans derived from 8 years of CALIOP and MODIS observations, *Atmospheric Chemistry and Physics*, *16*(5), 2877–2900, doi:10.5194/acp-16-2877-2016.

*Answers to Reviewer #1 Review on "Aerosol direct radiative effect over clouds from a synergy of OMI and MODIS reflectances" by de Graaf et al.*
The reviewer is thanked for the appreciation of the manuscript and the previous changes, which were very helpful and improved the manuscripts considerably.

Below the additional questioned are answered:

- The first question was about the all-sky DRE. This is something that is not considered in the manuscript. It should be clear from the title and the text from the beginning that only cloud scenes are considered in this manuscript. The all-sky DRE is complicated to derive from observations, since the clouds are very diverse and have very many effects on the DRE, even when 3D effects and cloud edge problems are not considered. The suggested computation by the reviewer is an approximation of the all-sky DRE that is only valid for homogenous cloud fields. It was now made even more clear in the manuscript that the derived DRE from OMI is valid only for sufficiently clouded scenes, where sufficiently was defined as FRESCO CF > 0.3, as stated in the text before. Also added was a warning that this should not be used to derive an all-sky DRE using the independent pixel approximation, as the OMI footprints are too large for that. The added section is:

*Note that the more general all-sky direct radiative effect of aerosols in both clear and cloudy scenes is often derived as* $\mathrm{DRE}_{\mathrm{all\ sky}} = f_{\mathrm{cld}} \cdot \mathrm{DRE}_{\mathrm{cld}} + (1 - f_{\mathrm{cld}}) \cdot \mathrm{DRE}_{\mathrm{clear}}$. *Here,* $\mathrm{DRE}_{\mathrm{cld}}$ *is the direct radiative effect of all aerosols in a completely overcast atmosphere,* $\mathrm{DRE}_{\mathrm{clear}}$ *the direct radiative effect of all aerosols in a cloud-free (Rayleigh) atmosphere, and* $f_{\mathrm{cld}}$ *is the fraction of clouds. However, the validity of this equation, known as the independent pixel approximation, is dependent on pixel size and cloud homogeneity. The cloud fraction* $f_{\mathrm{cld}}$ *is the fraction of an area where clouds appear with similar radiative properties. This may be true for satellites with sufficiently small pixels and homogenous cloud fields. However, in this paper the aerosol DRE is derived from OMI, which has a relatively large footprint. For OMI an* effective *cloud fraction is derived using the Fast Retrieval Scheme for Clouds from the Oxygen-A band (FRESCO) algorithm, using the $O_2$-$O_2$ absorption band at 477 nm and the DRE is derived for OMI pixels with an FRESCO CF > 0.3 to ensure sufficiently clouded scenes. The effective cloud fraction differs from the geometric cloud fraction, in that it is radiatively equivalent to the geometrial cloud fraction and cloud optical thickness of the scene, assuming complete cloud coverage. Therefore, COT and cloud droplet effective radius (CER) are retrieved assuming a completely clouded scene. Then, the aerosol DRE is computed using those cloud parameters again assuming complete cloud coverage. Although this is common for satellite cloud products, it should be understood that the OMI aerosol DRE dataset is not equivalent to the* $\mathrm{DRE}_{\mathrm{cld}}$. *A large part of the scenes with either small (geometrical) cloud fraction or small cloud optical thickness are not considered by selecting only scenes with FRESCO CF > 0.3. These scenes will have a small positive or negative aerosol DRE, as aerosol scattering dominates over dark surfaces. Therefore the average OMI aerosol DRE in this paper is higher than the average true cloud or all-sky aerosol DRE. However, the dataset can be used to validate simulations of the*

*aerosol DRE or other observational datasets where also scenes with CF > 0.3 are selected.*

- The second question is about the combination of OMI and MODIS reflectances. Again, the reviewer decomposes the reflectance in a cloudy and cloud-free part, adding them. However, radiatively this can only be done when the cloud is homogenous. In general, each cloud part with different COT and CER has to be added separtely. However, the discussion is not about how to add cloud-free and cloud reflectances. In general, the MODIS reflectances aggregated over the OMI footprint will yield a different average reflectance than the OMI reflectance for that footprint, due to differences in instrument response functions, uncertainties in viewing directions, calibration errors, changing scenes during the instrument overpasses, etc. Therefore, the reflectances have to be matched, which can be done since the spectra overlap. In general, the difference was found to be small, since both instruments are well calibrated. However, we found that especially for broken cloud fields the difference can be significant. Therefore, the physical significance of the 'scaling factor' is the uncertainty of the combination of observations, including all the effects mentioned above. However, the only significant effect is the change in reflectance due to cloud fraction difference. And this effect is already quantified in section 4.3 (Accuracy assessment) in the manuscript, where the accuracy of the cloud reflectance simulation is assessed. This uncertanty includes both the uncertainty of the measurement (combination) and the simulation.

- The references have been added.

- The aerosol DRE over clouds is defined in section 2 and valid for any selected OMI cloud pixel. It is not to be mistaken with any of the parameters defined by the reviewer.

- The same as above holds for the aerosol DRE for SCIAMACHY cloud pixels.

- Correct. This was changed.

- The acronym FRESCO was defined at first use.

*Answers to Reviewer # 2*

I reviewed a previous version of this manuscript. My main issues with the previous version were (1) discussion of the anisotropy factor B and related uncertainty; (2) choice of a threshold UVAI=0 as a baseline for the uncertainty calculations; and (3) the fairly simplistic nature of the 2016/2017 data analysis (mentioning ORACLES but not using the data). In this revision the authors have expanded the discussion of (1) and (2), which I appreciate, and added MODIS and OMI above-cloud satellite time series for (3) while noting that the comparison with ORACLES data is better suited for a separate paper (which I hope they do). This makes it more convincing, in my view, than the previous submission.

*The reviewer is thanked for the constructive criticism and helpful comments that have helped improve the manuscript. The remaining questions are answered below.*

As a result I do not have technical objections to the publication of this manuscript, although the other reviewer (Z. Zhang) is more of an expert in the forcing aspect than I am, so I would defer to their judgment.

I have a few minor comments, but otherwise find the manuscript acceptable for publication after technical corrections. I would be happy to review these corrections if the Editor feels it would be helpful, although I do not think it is necessary, provided the other reviewer is satisfied.

Previous comment on POLDER: the authors had cited a paper in preparation which compared the OMI/MODIS results against POLDER. I'd suggested the authors provide the results here or remove the reference, since we can't see the results otherwise (given it's a paper which has not been submitted yet). They replied that they have added the information and also removed the citation. It's not clear to me where the information about this comparison has been added, as it doesn't seem to be in the original section; there is a brief mention in section 4.2 of POLDER but that seems to be it? Mentioning just in case something was inadvertently omitted, but I think it is ok as-is.

*The POLDER reference has been added since it is now available as a discussion paper.*

Page 7 line 27: a reference for the MODIS sensor should be added (one is already provided for the other satellite instruments used). I don't have a particular strong feeling about which, but Salmonson et al (1989) is often used: https://ieeexplore.ieee.org/document/20292

*The reference was added.*

Page 10 line 2: Acronym FRESCO needs to be defined at first use.

*This was added.*

Page 21 line 4: Acronym AERONET needs to be defined at first use. Also, state which version you are using and provide a reference. It should be the current version 3, with citation Giles et al (2019): https://www.atmos-meas-tech.net/12/169/2019/amt-12-169-2019-discussion.html Also define the data level being used. I see this is level 1.5 rather than the standard level 2; after checking the AERONET website I see level 2 is not available yet for Ascension Island in 2017. It is worth mentioning the difference between levels and stating why level 1.5 is used here.

*The acronym was defined at first use. The used version was version 2, the level 1.5, for the reason mentioned by the reviewer. Version 3 is also available, but it showed rather different behaviour than the V2 data and was not used. This is mentioned in the manuscript and a reference was added.*

Page 22 lines 30-32: VIIRS and OMPS acronyms should be defined at first use. Also, there is more than just SNPP now, NOAA20 (formerly JPSS1) launched in late 2017.

*The acronyms was expanded and the NOAA20 reference added.*

Section 4.3.3 and more generally: in this section (and elsewhere) the authors say "error" often. I think a lot of these times, they really mean "uncertainty". For example, page 19 line 9 I think the authors mean the "uncertainty" in the DRE retrievals, not the error, since we don't have a truth to compare to. If possible it would also be good for the authors to clarify whether the estimates they provide in the paper refer to typical levels of uncertainty (e.g. 1-sigma), maximum likely uncertainty, or similar.

*Correct. The term error was replaced by 'uncertainty' where appropriate and total error by accuracy. The uncertainty estimates are specified.*